# ChromFound: Towards A Universal Foundation Model for Single-Cell Chromatin Accessibility Data

**Yifeng Jiao[1,2,*], Yuchen Liu[1,2,*], Yu Zhang[2], Xin Guo[1,2,†], Yushuai Wu[2], Chen Jiang[1,2],**
**Jiyang Li[2], Hongwei Zhang[1,2], Limei Han[1,2], Xin Gao[2,3,4,5], Yuan Qi[1,2,6,†], Yuan Cheng[1,2,†]**

[1]Artificial Intelligence Innovation and Incubation Institute, Fudan University
[2]Shanghai Academy of Artificial Intelligence for Science
[3]Computer, Electrical and Mathematical Sciences and Engineering Division, KAUST
[4]Center of Excellence for Smart Health, KAUST
[5]Center of Excellence on GenAI, KAUST
[6]Zhongshan Hospital, Fudan University
Correspondence to: {jiaoyifeng,guoxin}@sais.org.cn, {cheng_yuan,qiyuan}@fudan.edu.cn

## Abstract

The advent of single-cell Assay for Transposase-Accessible Chromatin using sequencing (scATAC-seq) offers an innovative perspective for deciphering regulatory mechanisms by assembling a vast repository of single-cell chromatin accessibility data. While foundation models have achieved significant success in single-cell transcriptomics, there is currently no foundation model for scATAC-seq that supports zero-shot high-quality cell identification and comprehensive multi-omics analysis simultaneously. Key challenges lie in the high dimensionality and sparsity of scATAC-seq data, as well as the lack of a standardized schema for representing open chromatin regions (OCRs). Here, we present **ChromFound**, a foundation model tailored for scATAC-seq. ChromFound utilizes a hybrid architecture and genome-aware tokenization to effectively capture genome-wide long contexts and regulatory signals from dynamic chromatin landscapes. Pretrained on 1.97 million cells from 30 tissues and 6 disease conditions, ChromFound demonstrates broad applicability across 6 diverse tasks. Notably, it achieves robust zero-shot performance in generating universal cell representations and exhibits excellent transferability in cell type annotation and cross-omics prediction. By uncovering enhancer-gene links undetected by existing computational methods, ChromFound offers a promising framework for understanding disease risk variants in the noncoding genome. The implementation of ChromFound is available via https://github.com/JohnsonKlose/ChromFound.

## 1 Introduction

The human genome harbors an extensive repository of open chromatin regions (OCRs) responsible for orchestrating precise spatial and temporal gene expression patterns [31]. Identifying OCRs at single-cell level is pivotal for understanding the regulatory landscape of individual cells, offering profound insights into cellular diversity and dynamic mechanisms of gene regulation [14, 10]. The single-cell Assay for Transposase-Accessible Chromatin using sequencing (scATAC-seq) [4] stands as the most prevalent genome-wide technique to enable the identification of OCRs at single-cell resolution. Its principal applications span cancer research, immunological studies, and neuroscience [52, 63, 61, 79].

---

*Equal contribution.
†Corresponding author.

39th Conference on Neural Information Processing Systems (NeurIPS 2025).

Furthermore, with initiatives such as the Human Cell Atlas [55] and other significant studies [17, 83], the scale of scATAC-seq data is expanding rapidly, opening new possibilities and presenting novel challenges in the exploration of OCRs at the same time.

Pretrained language models (PLMs) have achieved great success in deciphering the complex language of single-cell RNA sequencing (scRNA-seq) data [64, 11, 27, 77], validating their capabilities of fine-tuning for a variety of downstream tasks. Nevertheless, the development of generalizable foundation models for scATAC-seq remains under-explored, posing significant challenges for cellular representation learning and multi-task transfer. Here, we identify and discuss three major challenges.

(1). **scATAC-seq data are inherently high-dimensional and sparse**, with single cells spanning over millions of OCRs but typically less than 1% showing accessibility. This sparsity complicates the utilization of Transformer [67] adopted in transcriptomics [64, 11, 27, 70, 18, 77]. Other methods such as transforming accessibility into gene activity [22, 59], filtering highly variable OCRs [22, 62, 20], or binarizing accessibility [28, 13, 81] always cause substantial information loss [60, 43, 6].

(2). **scATAC-seq data are not formatted in a standardized feature space**. Dynamic chromatin landscapes across diverse sources and platforms make dictionary-based tokenization for genes and nucleotides unsuitable for OCRs. All deep learning models for scATAC-seq [73, 72, 12, 81] require dataset-specific training, leading to poor zero-shot performance.

(3). **current methods lack a unified integration of genomic information with chromatin accessibility profiles**. Both genomic information defining regulatory loci and chromatin accessibility profiles revealing their activity are essential in scATAC-seq data. However, VAE-based models [73, 72, 12] process accessibility matrices independently, while models like scBasset, CellSpace, and SANGO [81, 62, 82] predict binary accessibility from genomic sequences. These methods struggle to generalize across diverse downstream tasks.

To address these challenges, we present **ChromFound**, a universal foundation model for single-cell chromatin accessibility data. Its hybrid architecture integrates a Mamba block for efficient long-range sequence processing with a self-attention block to capture local regulatory dependencies within ±200 kb of transcriptional start sites, aligning with the typical range of enhancer-promoter interactions identified by Hi-C analysis [2]. ChromFound incorporates biologically informed OCR tokenization encoding genomic coordinates and non-binary accessibility profiles. This component ensures scalability and alignment across diverse scATAC-seq datasets from varied tissues, platforms, and protocols. Pretrained on over 1.97 million single cells from more than 30 organs or tissues and 6 major disease categories (e.g., Alzheimer's, Parkinson's, leukemia, glioma), ChromFound leverages 1.86 trillion training tokens, surpassing the scale of existing models such as Geneformer [64], scGPT [11], and scFoundation [27], all of which use fewer than 1 trillion tokens.

To validate the effectiveness of ChromFound, we conduct comprehensive evaluations across multiple tasks and datasets. In cell representation, ChromFound outperforms baselines instead of additional training, achieving an average ARI improvement of 17.02% across 8 scATAC-seq datasets from 4 tissues. Additional results demonstrate its robustness to denoise technical effect in zero-shot settings. For cell type annotation, ChromFound significantly improves accuracy and macro F1 score across all evaluated datasets and tissues. In cross-omics prediction, it accurately infers gene expression from chromatin accessibility, surpassing all baselines across 5 datasets. Furthermore, in the K562 cell line, ChromFound effectively identifies gene-enhancer relationships and regulatory perturbation effects, highlighting its potential for interpretable *cis*-regulatory modeling.

Our main contributions can be summarized as follows:

(1) ChromFound, the first scATAC-seq foundation model, employs a genome-wide architecture along with a genome-aware tokenization and is pretrained on a large corpus with 1.97 million single cells.

(2) ChromFound provides a strong zero-shot cell representation from an epigenetic perspective, as well as showing consistent robustness to data sparsity and batch effect.

(3) ChromFound shows great transfer learning capabilities in cell annotation and cross-omics prediction, validating biological significance in inferring enhancer-gene links and perturbation responses.

## 2  Related Work

Single-cell foundation models for scRNA-seq data have seen significant advancements, supporting tasks such as gene function prediction, cell annotation, and drug response modeling. Geneformer [64], scGPT [11], and scFoundation [27] are trained by large-scale datasets, while scBERT [77] and CellPLM [70] focus on specific objectives based on pre-training language models such as imputation and cell communications. However, existing models are predominantly tailored for scRNA-seq data and fail to model scATAC-seq data effectively. GET [18] achieves a level of predictive precision comparable to experimental replicates. It does so by integrating chromatin accessibility data and genomic sequence information in a pseudo-bulk format, rather than relying on single-cell resolution.

Deep learning has been widely applied to scATAC-seq data analysis for cell clustering and annotation. PeakVI [1] and PoissonVI [44] extend VAE-based frameworks, with the former disentangling biological from technical variation for batch correction and the latter introducing a Poisson likelihood to preserve quantitative accessibility. SCALE [73] and SCALEX [72] employ VAEs for latent representation learning and multi-omics integration, while CASTLE [12] enhances cell-type annotation via self-supervised learning. Beyond VAE-based methods, SnapATAC2 [84] scales to millions of cells using latent semantic indexing and spectral embedding, and Signac [59] provides a Seurat-based workflow for dimensionality reduction and multimodal integration. scBasset [81] links genomic sequences with accessibility profiles, and Cellcano [42] together with SANGO [82] employ two-step MLPs for cell-type annotation. Despite their advances, these models remain limited by architectural constraints and data scarcity, restricting scalability and generalization.

## 3  Methods

### 3.1  OCR Tokenization

To encode chromatin accessibility data, we treat each OCR as a token by three essential components: chromosome embedding, positional embedding for genomic coordinates, and continuous accessibility embedding. The tokenization integrates these components to form a comprehensive representation of each OCR. Our unified token representation is designed to address two fundamental challenges. First, by encoding start and end genomic coordinates through sinusoidal positional embeddings, ChromFound captures the full extent of each OCR, accommodating varying fragment lengths. Second, by avoiding reliance on a fixed OCR vocabulary, ChromFound flexibly represents tissue-specific or novel OCRs, thereby mitigating misalignment and dropout across tissues. In the following, we describe the encoding strategy for each component in detail.

**Chromosome Embedding**: Chromosome identity is encoded using a learnable embedding lookup table. The size of the chromosome embedding lookup table is 25, with 24 corresponding to the number of human chromosomes and 1 designated for the padding token. We define the learnable embedding matrix $\mathbf{W}_c \in \mathbb{R}^{|\mathcal{C}| \times d_{\text{model}}}$ and the chromosome index $c_j$ of OCR $j$. ChromFound maps the $c_j$ to $\mathbf{E}_{c,j}$ by the following equation:

$$\mathbf{E}_{c,j} = \mathbf{W}_c[c_j, :].  \tag{1}$$

**Positional Embedding for Genomic Coordinates**: Since some studies [36, 76] have noted that state space models inherently handle sequential information through their recurrent nature, we use positional embedding to encode genomic coordinates instead of the sequence order. Specifically, the embedding of the starting position and the ending position for the OCR $j$ is computed as:

$$\mathbf{E}_{p,j}^s = \begin{bmatrix} \sin\left(\frac{p_j^s}{\text{temp} \cdot 10000^{2k/d_{\text{model}}}}\right) \\ \cos\left(\frac{p_j^s}{\text{temp} \cdot 10000^{2k/d_{\text{model}}}}\right) \end{bmatrix}, \quad \mathbf{E}_{p,j}^e = \begin{bmatrix} \sin\left(\frac{p_j^e}{\text{temp} \cdot 10000^{2k/d_{\text{model}}}}\right) \\ \cos\left(\frac{p_j^e}{\text{temp} \cdot 10000^{2k/d_{\text{model}}}}\right) \end{bmatrix}  \tag{2}$$

where $p_j^s$ and $p_j^e$ represent the starting and end position for the OCR $j$ on the GRCh38 reference genome, temp is a hyperparameter set to 100000 in ChromFound, and $d_{\text{model}}$ is the dimension of the embedding layer. The index $k$ spans $\{0, 1, \ldots, d_{\text{model}}/2\}$, with sine and cosine functions alternately assigned to even and odd dimensions of the embedding vector.

**Accessibility Embedding**: The accessibility value for the $j$th OCR, denoted as $X_j$, is a continuous scalar processed by the procedures detailed in Appendix D. To project the value $X_j$ into the hidden dimension, we use a linear transformation as $\mathbf{E}_{a,j} = \text{Linear}(X_j)$.

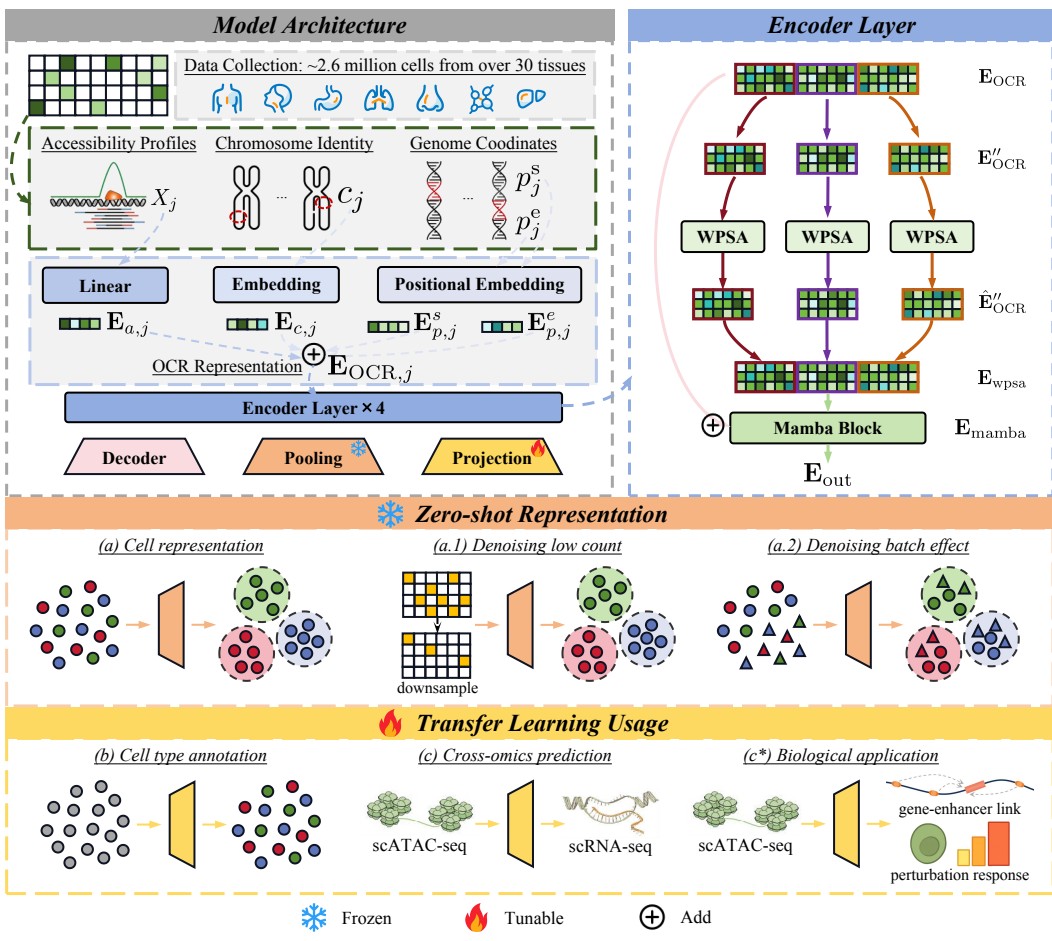

Figure 1: **An overview of ChromFound architecture**. The methods of ChromFound are listed in Section 3. OCR tokenization are described in Section 3.1. The hybrid encoder layer is described in Section 3.2. Other pretraining details of datasets and implementation are included in Section 3.3. All evaluation results are detailed in Section 4.

**OCR Representation**: The unified representation $\mathbf{E}_{\text{OCR},j} \in \mathbb{R}^{1 \times D}$ of the $j$th OCR is obtained by adding the previous embeddings together:

$$\mathbf{E}_{\text{OCR},j} = \mathbf{E}_{c,j} + \mathbf{E}_{p,j}^s + \mathbf{E}_{p,j}^e + \mathbf{E}_{a,j}. \tag{3}$$

## 3.2 Hybrid Encoder Layer

The ChromFound encoder layer transforms the OCR representation $\mathbf{E}_{\text{OCR}} \in \mathbb{R}^{L \times D}$ into a latent space, where $L$ denotes the sequence length and $D$ is the embedding dimension. As illustrated in Fig. 1, ChromFound incorporates a self-attention module (Section 3.2.1) to effectively capture local dependencies among OCRs, and integrates a Mamba layer (Section 3.2.2) to handle ultra-long OCR sequences. This architecture is motivated by biological insights, recognizing that both short- and long-range interactions among OCRs are critical to gene regulatory mechanisms [30, 51, 54].

### 3.2.1 Window Partition Self-Attention (WPSA)

Due to its computational complexity $O(L^2)$, the vanilla attention mechanism [67] is not directly applicable to scATAC-seq data containing millions of OCRs. To address these challenges, ChromFound introduces an optimized self-attention layer named **W**indow **P**artition **S**elf-**A**ttention (WPSA), which applies the attention mechanism within local windows of size $W$, where $W \ll L$.

**Window Partition**: Given the OCR representation $\mathbf{E}_{\text{OCR}} \in \mathbb{R}^{L \times D}$, $L$ may not be a multiple of $W$. We define the number of windows $N = \lceil \frac{L}{W} \rceil$ and pad $\mathbf{E}_{\text{OCR}}$ with zeros up to $N \times W$ along the dimension $L$. As a result, the padded sequence $\mathbf{E}'_{\text{OCR}} \in \mathbb{R}^{(N \cdot W) \times D}$ is reshaped as $\mathbf{E}''_{\text{OCR}} \in \mathbb{R}^{N \times W \times D}$.

**Self-Attention**: In practice, we set the window size to $W = 256$, which approximately spans 200 kb upstream and downstream of transcriptional start site (TSS). This setting is consistent with the scope of enhancer-promoter interactions suggested by high-resolution Hi-C analysis [2].

**Window Restore**: After applying self-attention within each window, we merge outputs of each window $\hat{\mathbf{E}}''_{\text{OCR}}$ back into $\hat{\mathbf{E}}_{\text{wpsa}}$:

$$\hat{\mathbf{E}}_{\text{wpsa}} = \text{concat}\left[\hat{\mathbf{E}}''_{\text{OCR},1}, \ldots, \hat{\mathbf{E}}''_{\text{OCR},N}\right] \in \mathbb{R}^{(N \cdot W) \times D}. \tag{4}$$

Furthermore, we remove all zero padding tokens in $\hat{\mathbf{E}}_{\text{wpsa}}$ and get the WPSA output $\mathbf{E}_{\text{wpsa}} \in \mathbb{R}^{L \times D}$.

### 3.2.2 Mamba Block

Mamba [24] is capable of processing extended sequences due to the linear-time complexity of state space models (SSMs) [26, 25], making it particularly well suited for modeling high-resolution scATAC-seq data. When combined with the WPSA module, which amplifies local signals within predefined windows, the Mamba block enables effective integration of both short- and long-range dependencies among OCRs across the genome. This hybrid architecture mitigates the degradation of long-range information typically caused by relying solely on zero-order discretization [50, 80]. To reduce parameter count and refine representations, we project $\mathbf{E}_{\text{wpsa}}$ to a lower-dimensional space $\mathbb{R}^{L \times D_{\text{low}}}$ and then back, which is defined as:

$$\mathbf{E}_{\text{down}} = \mathbf{E}_{\text{wpsa}} W_{\text{down}}, \quad \mathbf{E}_{\text{mamba}} = \text{Mamba}(\mathbf{E}_{\text{down}}), \quad \mathbf{E}_{\text{up}} = \mathbf{E}_{\text{mamba}} W_{\text{up}}, \tag{5}$$

where $W_{\text{down}} \in \mathbb{R}^{D \times D_{\text{low}}}$ ($D_{\text{low}} < D$) and $W_{\text{up}} \in \mathbb{R}^{D_{\text{low}} \times D}$.

### 3.2.3 Overall Workflow

The overall workflow of the encoder layer is illustrated in Fig. 1 and proceeded as:

$$\mathbf{E}_{\text{out}} = \mathbf{E}_{\text{OCR}} + W_{\text{up}} \cdot \text{Mamba}(W_{\text{down}} \cdot \text{WPSA}(\text{RMSNorm}(\mathbf{E}_{\text{OCR}}))) \tag{6}$$

### 3.3 Pre-training

**Datasets**: We assemble a large-scale scATAC-seq dataset comprising over 2.64 million cells detailed in Table 6. For model pretraining, we select a representative subset of 1.97 million cells from more than 30 distinct organs and tissues. To rigorously evaluate the generalization ability of ChromFound, we additionally compile a benchmark dataset containing 0.67 million cells from diverse sources, including tissues such as bone and retina excluded from pretraining.

**Implementation**: ChromFound is trained for 5 epochs over 80 hours on a compute cluster comprising 4 machines with a total of 32 NVIDIA A100 GPUs. The effective batch size is set to 128. We use the AdamW optimizer [39] with a maximum learning rate of $5 \times 10^{-5}$. The embedding dimension and the hidden size of WPSA $D$ are set to 128, while dimension $D_{\text{low}}$ of the Mamba block is set to 32. The model architecture consists of 4 stacked encoder layers in total.

**Training objective**: Due to the high sparsity of scATAC-seq data, ChromFound adopts a masking strategy that simultaneously targets both zero and non-zero OCR values. Inspired by xTrimoGene [21], the setting encourages the model to predict non-binary chromatin accessibility across the entire profile, mitigating the risk of learning representations dominated by zero entries. Poisson/ZINB losses often cause unstable optimization due to gradients dominated by nonzero entries. Instead, mean squared error (MSE) loss on log-transformed, normalized signals ensures stability. The MLP layer is employed as the decoder to reconstruct the original input $X_i$. The reconstruction output of cell $i$ is denoted as $\hat{X}_i$. The pretraining objective is the MSE loss computed over the masked positions $\mathcal{M}_i$ in each cell $i$:

$$\mathcal{L}_{mse,i} = \frac{1}{|\mathcal{M}_i|} \sum_{i \in \mathcal{M}_i} \left(X_i - \hat{X}_i\right)^2. \tag{7}$$

Table 1: Results of cell clustering tasks. Result style: **best**, second best, **relative gains**.

| Dataset | Tissue | Model | ARI(↑) | FMI(↑) | NMI(↑) | AMI(↑) |
|---|---|---|---|---|---|---|
| Morabito130K [48]
(batch 1) | Cortex | PeakVI [1] | 0.4558±0.0038 | 0.6286±0.0022 | 0.6964±0.0012 | 0.6940±0.0012 |
| | | PoissonVI [44] | 0.3673±0.0028 | 0.5673±0.0018 | 0.5878±0.0001 | 0.5845±0.0001 |
| | | SnapATAC2 [84] | 0.5118±0.0315 | 0.6740±0.0145 | 0.7202±0.0041 | 0.7179±0.0042 |
| | | Signac [59] | 0.5013±0.0507 | 0.7035±0.0177 | 0.5580±0.0160 | 0.5534±0.0165 |
| | | SCALE [73] | 0.5460±0.0074 | 0.6896±0.0042 | 0.7282±0.0016 | 0.7275±0.0016 |
| | | SCALEX [72] | 0.4930±0.0022 | 0.6505±0.0013 | 0.6958±0.0001 | 0.6949±0.0001 |
| | | CASTLE [12] | 0.4425±0.0029 | 0.6109±0.0019 | 0.7086±0.0012 | 0.7077±0.0012 |
| | | scBasset [81] | 0.5039±0.0039 | 0.6747±0.0022 | 0.6674±0.0005 | 0.6647±0.0005 |
| | | **ChromFound** | **0.6890±0.0387** | **0.7943±0.0144** | **0.7779±0.0037** | **0.7760±0.0038** |
| | | Gains(%) | **26.20** | **15.17** | **6.81** | **6.66** |
| Morabito130K [48]
(batch 2) | Cortex | PeakVI [1] | 0.5512±0.0016 | 0.6394±0.0032 | 0.7050±0.0006 | 0.7028±0.0006 |
| | | PoissonVI [44] | 0.5334±0.0052 | 0.6388±0.0033 | 0.6004±0.0001 | 0.5983±0.0001 |
| | | SnapATAC2 [84] | 0.5325±0.0072 | 0.6380±0.0045 | 0.7038±0.0018 | 0.7015±0.0018 |
| | | Signac [59] | 0.5069±0.0123 | 0.6478±0.0064 | 0.5888±0.0032 | 0.5851±0.0033 |
| | | SCALE [73] | 0.5306±0.0048 | 0.6394±0.0032 | 0.6221±0.0007 | 0.6200±0.0007 |
| | | SCALEX [72] | 0.3905±0.0041 | 0.5252±0.0027 | 0.5361±0.0028 | 0.5336±0.0029 |
| | | CASTLE [12] | 0.3743±0.0016 | 0.5087±0.0010 | 0.4636±0.0021 | 0.4608±0.0022 |
| | | scBasset [81] | 0.5373±0.0024 | 0.6424±0.0015 | 0.6659±0.0003 | 0.6641±0.0003 |
| | | **ChromFound** | **0.6278±0.0043** | **0.7156±0.0025** | **0.7321±0.0016** | **0.7299±0.0016** |
| | | Gains(%) | **16.83** | **10.47** | **3.84** | **3.86** |
| Kuppe139K [33]
(donar av3) | Heart | PeakVI [1] | 0.4159±0.0007 | 0.5434±0.0006 | 0.6024±0.0005 | 0.6000±0.0006 |
| | | PoissonVI [44] | 0.3969±0.0005 | 0.5232±0.0004 | 0.5717±0.0001 | 0.5688±0.0001 |
| | | SnapATAC2 [84] | 0.4571±0.0099 | 0.5564±0.0064 | 0.6625±0.0013 | 0.6502±0.0013 |
| | | Signac [59] | 0.3123±0.0047 | 0.4601±0.0028 | 0.5272±0.0028 | 0.5238±0.0029 |
| | | SCALE [73] | 0.4475±0.0015 | 0.5654±0.0011 | 0.6005±0.0003 | 0.5978±0.0003 |
| | | SCALEX [72] | 0.3819±0.0007 | 0.5061±0.0005 | 0.4829±0.0000 | 0.4797±0.0000 |
| | | CASTLE [12] | 0.3886±0.0017 | 0.5118±0.0110 | 0.4765±0.0001 | 0.4729±0.0001 |
| | | scBasset [81] | 0.4553±0.0022 | 0.5729±0.0016 | 0.6539±0.0006 | 0.6516±0.0006 |
| | | **ChromFound** | **0.5828±0.0052** | **0.6757±0.0034** | **0.7207±0.0008** | **0.7187±0.0008** |
| | | Gains(%) | **27.75** | **17.94** | **8.78** | **10.31** |
| Kuppe139K [33]
(donar av10) | Heart | PeakVI [1] | 0.4301±0.0030 | 0.6383±0.0021 | 0.6149±0.0022 | 0.6144±0.0022 |
| | | PoissonVI [44] | 0.3606±0.0054 | 0.5929±0.0027 | 0.4472±0.0066 | 0.4464±0.0066 |
| | | SnapATAC2 [84] | 0.5515±0.0312 | 0.7285±0.0124 | 0.6867±0.0060 | 0.6863±0.0060 |
| | | Signac [59] | 0.1159±0.0112 | 0.5093±0.0053 | 0.2294±0.0076 | 0.2281±0.0076 |
| | | SCALE [73] | 0.4517±0.0087 | 0.6596±0.0040 | 0.5397±0.0041 | 0.5391±0.0041 |
| | | SCALEX [72] | 0.4742±0.0105 | 0.6770±0.0055 | 0.5040±0.0079 | 0.5033±0.0080 |
| | | CASTLE [12] | 0.4739±0.0000 | 0.6786±0.0000 | 0.5204±0.0000 | 0.5197±0.0000 |
| | | scBasset [81] | 0.5823±0.0111 | 0.7459±0.0050 | 0.6780±0.0001 | 0.6776±0.0001 |
| | | **ChromFound** | **0.6774±0.0445** | **0.8109±0.0169** | **0.7369±0.0137** | **0.7365±0.0138** |
| | | Gains(%) | **16.34** | **8.71** | **7.31** | **7.31** |
| Liang154K [35]
(sample D026_13) | Retina | PeakVI [1] | 0.2885±0.0024 | 0.5543±0.0019 | 0.5644±0.0019 | 0.5638±0.0019 |
| | | PoissonVI [44] | 0.3162±0.0025 | 0.5766±0.0017 | 0.6081±0.0017 | 0.6076±0.0017 |
| | | SnapATAC2 [84] | 0.5606±0.0569 | 0.7207±0.0199 | 0.7045±0.0085 | 0.7041±0.0085 |
| | | Signac [59] | 0.5538±0.1051 | 0.7451±0.0095 | 0.4931±0.0535 | 0.4919±0.0538 |
| | | SCALE [73] | 0.5512±0.0386 | 0.7285±0.0075 | 0.7172±0.0051 | 0.7168±0.0051 |
| | | SCALEX [72] | 0.5069±0.0048 | 0.7241±0.0022 | 0.6759±0.0005 | 0.6755±0.0005 |
| | | CASTLE [12] | 0.2424±0.0032 | 0.5136±0.0022 | 0.5298±0.0033 | 0.5292±0.0033 |
| | | scBasset [81] | 0.4330±0.0049 | 0.6699±0.0029 | 0.6903±0.0015 | 0.6898±0.0015 |
| | | **ChromFound** | **0.6668±0.0500** | **0.8149±0.0169** | **0.7644±0.0093** | **0.7641±0.0093** |
| | | Gains(%) | **18.94** | **9.37** | **6.59** | **6.60** |
| Liang154K [35]
(sample D19D008) | Retina | PeakVI [1] | 0.4702±0.0061 | 0.6318±0.0037 | 0.7229±0.0024 | 0.7224±0.0025 |
| | | PoissonVI [44] | 0.4978±0.0122 | 0.6527±0.0065 | 0.7400±0.0028 | 0.7395±0.0028 |
| | | SnapATAC2 [84] | 0.5814±0.0019 | 0.7173±0.0010 | 0.7881±0.0011 | 0.7877±0.0011 |
| | | Signac [59] | 0.5146±0.0532 | 0.7339±0.0119 | 0.5235±0.0307 | 0.5222±0.0309 |
| | | SCALE [73] | 0.6129±0.0131 | 0.7465±0.0113 | 0.7981±0.0020 | 0.7993±0.0058 |
| | | SCALEX [72] | 0.5424±0.0041 | 0.6872±0.0023 | 0.6976±0.0019 | 0.6971±0.0019 |
| | | CASTLE [12] | 0.4272±0.0052 | 0.5982±0.0034 | 0.6614±0.0034 | 0.6608±0.0034 |
| | | scBasset [81] | 0.6027±0.0139 | 0.7313±0.0066 | 0.7881±0.0010 | 0.7878±0.0010 |
| | | **ChromFound** | **0.6688±0.0264** | **0.7767±0.0123** | **0.8183±0.0032** | **0.8179±0.0032** |
| | | Gains(%) | **9.12** | **4.05** | **2.53** | **2.34** |
| PBMC169K [15]
(batch VIB_10xv1_1) | PBMC | PeakVI [1] | 0.5417±0.0022 | 0.6340±0.0015 | 0.6639±0.0009 | 0.6627±0.0009 |
| | | PoissonVI [44] | 0.6314±0.0027 | 0.7073±0.0018 | 0.7353±0.0012 | 0.7343±0.0012 |
| | | SnapATAC2 [84] | 0.6246±0.0112 | 0.7085±0.0067 | 0.7551±0.0023 | 0.7541±0.0023 |
| | | Signac [59] | 0.1832±0.0136 | 0.4543±0.0043 | 0.3064±0.0198 | 0.3027±0.0201 |
| | | SCALE [73] | 0.6158±0.0048 | 0.6959±0.0031 | 0.7632±0.0010 | 0.7623±0.0010 |
| | | SCALEX [72] | 0.6121±0.0048 | 0.6927±0.0015 | 0.7593±0.0008 | 0.7584±0.0008 |
| | | CASTLE [12] | 0.5530±0.0018 | 0.6439±0.0012 | 0.7263±0.0004 | 0.7252±0.0004 |
| | | scBasset [81] | 0.6240±0.0001 | 0.7019±0.0001 | 0.7660±0.0000 | 0.7651±0.0000 |
| | | **ChromFound** | **0.6953±0.0035** | **0.7601±0.0023** | **0.7860±0.0007** | **0.7852±0.0008** |
| | | Gains(%) | **10.12** | **7.28** | **2.61** | **2.62** |
| PBMC169K [15]
(batch BIO_ddseq_1) | PBMC | PeakVI [1] | 0.2915±0.0005 | 0.5041±0.0004 | 0.4623±0.0004 | 0.4615±0.0005 |
| | | PoissonVI [44] | 0.4103±0.0045 | 0.5994±0.0028 | 0.5656±0.0035 | 0.5649±0.0035 |
| | | SnapATAC2 [84] | 0.4126±0.0318 | 0.6149±0.0149 | 0.5761±0.0071 | 0.5754±0.0071 |
| | | Signac [59] | 0.1885±0.0097 | 0.5249±0.0056 | 0.2852±0.0040 | 0.2839±0.0040 |
| | | SCALE [73] | 0.4359±0.0181 | 0.6250±0.0101 | 0.5594±0.0045 | 0.5587±0.0045 |
| | | SCALEX [72] | 0.4087±0.0197 | 0.6063±0.0106 | 0.5519±0.0066 | 0.5512±0.0066 |
| | | CASTLE [12] | 0.3504±0.0042 | 0.5555±0.0026 | 0.5512±0.0013 | 0.5506±0.0013 |
| | | scBasset [81] | 0.4362±0.0012 | 0.6223±0.0008 | 0.5577±0.0008 | 0.5571±0.0008 |
| | | **ChromFound** | **0.4835±0.0082** | **0.6604±0.0042** | **0.5950±0.0036** | **0.5944±0.0036** |
| | | Gains(%) | **10.84** | **5.66** | **3.28** | **3.30** |

# 4 Experiments

To systematically evaluate the performance of ChromFound, we conduct experiments on various tissues and datasets. We first assess its zero-shot representation capabilities, demonstrating strong robustness to both data sparsity and batch effects. We then evaluate the transferability in downstream tasks, including cell type annotation and cross-omics prediction. Finally, we validate the biological utility of ChromFound by accurately inferring gene-enhancer regulatory links and gene expression responses to enhancer perturbations. All scATAC-seq datasets used in experiments are preprocessed according to the procedures detailed in Appendix D.

## 4.1 Cell representation

ChromFound generates low-dimensional cell representations by applying PCA to the encoder outputs after the pooling layer. Details of the implementations are provided in Appendix B.1. For comparison, we include scBasset [81] and three VAE-based models [73, 72, 12], which require self-supervised learning on each specific dataset. We conduct experiments on eight datasets from four tissue types and evaluate performance using four clustering metrics to ensure a fair and reliable comparison.

All results presented in Table 1 reveal that ChromFound outperforms existing SOTA methods in all tissues and metrics, with average improvements of 17.02%, 10.39%, 6.72% and 6.69% in ARI, FMI, NMI and AMI, respectively. The greater improvements observed in ARI compared to NMI and AMI infer the robustness of ChromFound to technical noise such as low read depth and batch effects, as ARI is particularly sensitive to local structures within the cell representation space. We discuss ChromFound's ability to denoise low count and batch effect as below.

**Denoising low count**: Some studies [6, 34, 62] have acknowledged that the prevalence of missing signals in scATAC-seq makes standard analyses challenging. Although all datasets in Table 1 already exhibit high sparsity (~99%), we further simulate increasingly sparse conditions by downsampling accessible OCRs to 10%, 20%, 30%, 40% and 50% of their original counts. We adopt the same benchmarking methods and clustering metrics as listed in Table 1. As shown in Fig. 2, ChromFound consistently maintains stable performance across all levels of downsampling, with its advantage over baseline methods becoming increasingly pronounced as the sparsity intensifies. Furthermore, we observe that ChromFound performs better on the retina dataset under stronger downsampling. This improvement likely reflects protocol-specific artifacts, as retina data contain more frequently accessible OCRs, introducing redundancy and noise. Random downsampling suppresses these excessive signals, enhances the signal-to-noise ratio, and yields the observed performance gain.

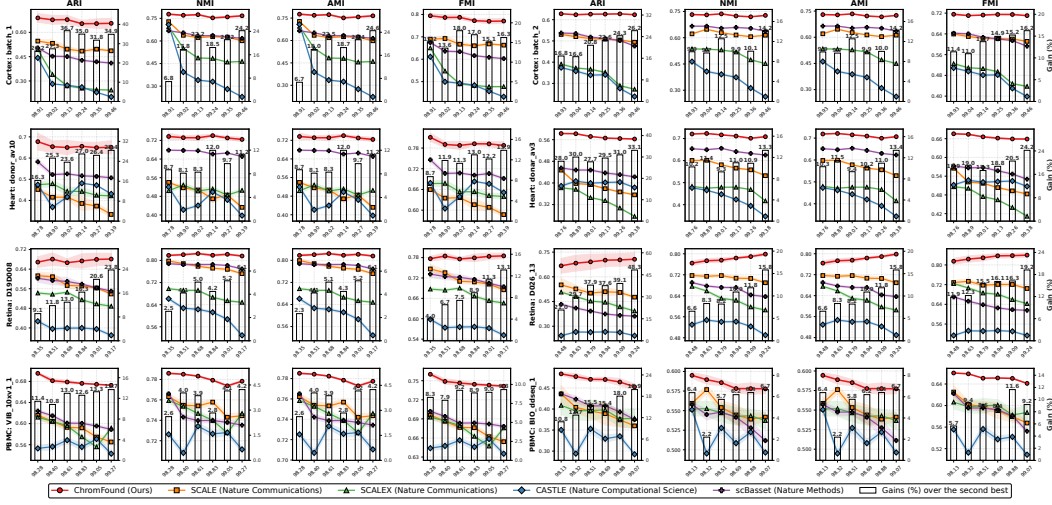

Figure 2: Results of denoising low count. The left y-axis shows the absolute metric values (scatter plot) and the right y-axis indicates the relative gains (%) over the second-best methods (bar plot).

**Denoising batch effect**: Denoising batch effects remains a major challenge in scATAC-seq analysis [46, 62, 41]. We evaluate ChromFound on datasets from 4 tissues using biological conservation metrics [41] consistent with scGPT [11]. In addition to PeakVI [1], PoissonVI [44], SCALEX [72] and scBasset [81], we include five specialized batch integration methods [38, 29, 32, 74, 37] for comparison. As shown in Table 2, ChromFound in zero-shot settings outperforms existing methods by more effectively mitigating batch effects while better preserving biologically significant variation.

Table 2: Results of denoising batch effect. Result style: **best**, second best, **relative gains**. AvgBIO is the average of $ARI_{cell}$, $NMI_{cell}$ and $ASW_{cell}$ measuring biological consistency. AvgBATCH is computed as the average of $ASW_{batch}$ and GraphConn to summarize the batch mixing performance.

| Model | **Bone** To326K [65] batch 43/44 | | **Heart** Kuppe139K [33] donar av3/av10 | | **PBMC** 169K [15] HAR ddseq 1/2 | | **Cortex** Morabito130K [48] batch 1/2 | |
|---|---|---|---|---|---|---|---|---|
| | AvgBIO(↑) | AvgBATCH(↑) | AvgBIO(↑) | AvgBATCH(↑) | AvgBIO(↑) | AvgBATCH(↑) | AvgBIO(↑) | AvgBATCH(↑) |
| scVI [38] | 0.3713 | 0.9026 | 0.7303 | 0.8448 | 0.6101 | 0.7932 | 0.7238 | 0.9217 |
| PeakVI [1] | 0.2981 | 0.9173 | 0.5657 | 0.7564 | 0.6172 | 0.7889 | 0.7256 | 0.9254 |
| PoissonVI [44] | 0.2936 | 0.9178 | 0.5621 | 0.7719 | 0.6029 | 0.8081 | 0.7240 | 0.9279 |
| Scanorama [29] | 0.4386 | 0.8722 | 0.5320 | 0.7779 | 0.6036 | 0.7964 | 0.6895 | 0.9243 |
| Harmony [32] | 0.3789 | 0.8558 | 0.5119 | 0.8390 | 0.6232 | 0.7923 | 0.7152 | 0.9101 |
| scANVI [74] | 0.4351 | 0.8479 | 0.6810 | 0.8401 | 0.6028 | 0.8035 | 0.7185 | 0.9208 |
| Liger [37] | 0.1992 | 0.6471 | 0.7286 | 0.8019 | 0.6051 | 0.8009 | 0.6822 | 0.9208 |
| SCALEX [72] | 0.1778 | 0.9177 | 0.7596 | 0.8415 | 0.5989 | 0.8004 | 0.4789 | 0.8728 |
| scBasset [81] | 0.3650 | 0.9208 | 0.6216 | 0.8591 | 0.6024 | 0.8014 | 0.7045 | 0.9301 |
| **ChromFound** | **0.6408** | **0.9289** | **0.8180** | **0.8679** | **0.6443** | **0.8217** | **0.7440** | **0.9565** |
| Gain(%) | 46.07 | 0.88 | 7.69 | 1.03 | 3.37 | 2.25 | 2.78 | 2.84 |

## 4.2 Cell Type Annotation

Computational cell type identification is a fundamental task in single-cell omics analysis. To adapt ChromFound for this task, we introduce a feature projection head followed by MLP layers as described in Appendix B.2. Our methods are compared to three strong baseline models designed for cell type annotation of scATAC-seq, Cellcano [42], EpiAnno [7] and SANGO [82]. The variant of ChromFound trained from scratch is also included to assess the contribution of pretraining. Fig. 3 highlights that ChromFound consistently outperforms all benchmark methods, achieving the highest performance in macro F1 score and demonstrating its ability to accurately annotate diverse cell types. The confusion matrices on PBMC169K [15] for ChromFound and CellCano [42] are shown in Fig. 9. The results indicate that Cellcano struggles to correctly classify rare cell types such as natural killer cells, CD16$^+$ monocytes, and dendritic cells, likely due to their low abundance. In contrast, ChromFound significantly improves the classification accuracy for these cell types, enabling a more precise understanding of immune cell composition and dynamics.

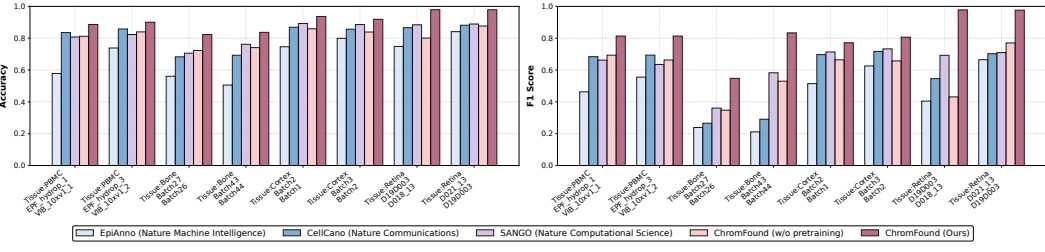

Figure 3: Results of cell type annotation, evaluated by Accuracy (left) and macro F1 Score (right).

## 4.3 Cross-omics Prediction

Although recent advances have enabled the simultaneous profiling of multiple assays, most single-cell datasets remain isolated, posing significant challenges for effective multi-omics analysis. To explore the potential of ChromFound in multi-omics prediction, we utilize scATAC-seq data as the source modality and scRNA-seq data as the target modality. Implementation details are provided in Appendix B.3. For comprehensive evaluation, we benchmark on five paired scATAC-seq and scRNA-seq datasets [86, 65, 40, 5, 87] using two standard correlation metrics. Three representative baseline

methods [71, 78, 69] are included with default configurations from the DANCE toolkit [16]. As shown in Table 3, ChromFound consistently outperforms all baselines across datasets in both PCC and CCC, highlighting its strong capability in cross-omics prediction. These results highlight the potential of ChromFound to facilitate biological analyses such as inferring gene-enhancer relationships and predicting transcriptional response to enhancer perturbation. The two biological applications are discussed in detail below.

Table 3: Cross-omics prediction results. Result style: **best**, second best, **relative gains**. ChromFound* stands for the version of ChromFound without loading the pretrained checkpoint.

| Model | Cortex Zhu45K [86] | | Bone To326K [65] | | BMMC multiome 2021 [40] | | BMMC atac2gex 2022 [5] | | Cell lines Zhu11K [87] | |
|---|---|---|---|---|---|---|---|---|---|---|
| | PCC(↑) | CCC(↑) | PCC(↑) | CCC(↑) | PCC(↑) | CCC(↑) | PCC↑ | CCC(↑) | PCC(↑) | CCC(↑) |
| BABEL [71] | 0.7975 | 0.7687 | 0.8020 | 0.7223 | 0.3854 | 0.3695 | 0.8901 | 0.8329 | 0.9196 | 0.8608 |
| CMAE [78] | 0.7973 | 0.7525 | 0.7986 | 0.6683 | 0.3435 | 0.3204 | 0.8983 | 0.7990 | 0.9136 | 0.8539 |
| scMM [47] | 0.7793 | 0.7325 | 0.7586 | 0.6513 | 0.3718 | 0.2983 | 0.8321 | 0.7027 | 0.8587 | 0.8062 |
| scMoGNN [69] | 0.8001 | 0.7210 | 0.7957 | 0.7233 | 0.4168 | 0.3911 | 0.9124 | 0.8655 | 0.7976 | 0.7263 |
| ChromFound* | 0.7864 | 0.7674 | 0.7887 | 0.6919 | 0.4213 | 0.3989 | 0.9279 | 0.8704 | 0.8992 | 0.8501 |
| **ChromFound** | **0.8316** | **0.8064** | **0.8304** | **0.7472** | **0.4249** | **0.4032** | **0.9293** | **0.8818** | **0.9449** | **0.9071** |
| **Gain(%)** | **4.10** | **4.67** | **3.54** | **3.30** | **1.94** | **3.09** | **1.85** | **1.88** | **2.67** | **5.11** |

**Biological Application: Predicting enhancer-gene link and perturbation response**

Enhancer elements in the human genome harbor thousands of genetic variants that regulate gene expression and contribute to the risk of common diseases [85]. In our approach detailed in Appendix B.4, we simulate OCR knockdown and estimate the resulting changes in gene expression. The absolute change reflects the likelihood of an enhancer-gene regulatory link, while the relative change captures the direction and strength of the transcriptional response. We perform experiments on 141 K562 cells from the test split in Zhu11K [87], using the cross-omics model trained as evaluated in Table 3. Fulco4K [87] provides the ground truth of enhancer-gene links and quantitative effects through CRISPRi perturbations in K562 human erythroleukemia cells. For clarity, the evaluation focuses on the COPZ1 and HNRNPA1 genes, which are involved in cancer-related processes such as stress adaptation and RNA splicing regulation [58, 8]. BABEL [71] and CMAE [78] are included as benchmark methods due to their strong performance in cross-omics prediction in Zhu11K [87]. As shown in Fig. 4(a), ChromFound achieves the highest AUC scores with 0.77 for COPZ1 and 0.61 for HNRNPA1, accurately identifying enhancer-gene links. Fig. 4(b) further shows that ChromFound better predicts transcriptional responses to enhancer perturbations. In contrast, BABEL [71] and CMAE [78] filter for the top 10,000 highly variable OCRs [16], leading to zero predictions of most enhancer perturbations. This limitation is evident in both Fig.4(a) and Fig.4(b), highlighting the advantage of ChromFound's genome-wide modeling to support a more comprehensive understanding of regulatory mechanisms.

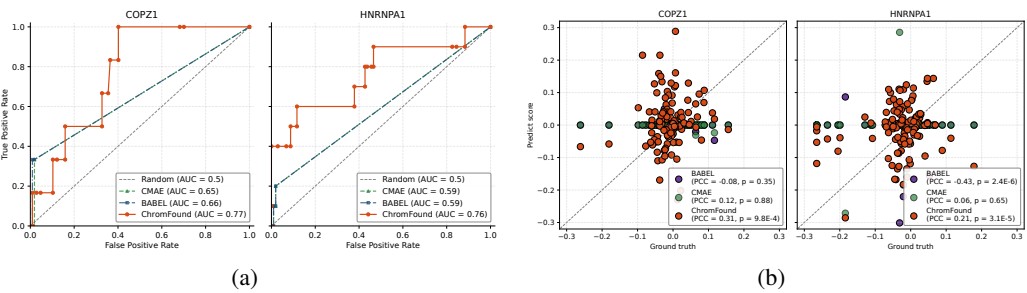

Figure 4: **(a)** ROC curves illustrate the performance of enhancer-gene link prediction, where true positive rates are plotted against false positive rates across varying prediction thresholds. Ground truth labels are defined based on the significance (FDR < 0.05) of expression changes after enhancer perturbation. **(b)** Scatter plots represent a specific enhancer perturbation. The y-axis shows the predicted scores of average gene responses of enhancer perturbation, while the x-axis represents the real effects of post-perturbation. Both magnitudes are rescaled to the unit norm.

## 4.4 Ablation study

In this section, we conduct an ablation study to address three questions using the PBMC169K [15], sampled from batch VIB_10xv1_1. The results of cell representation are summarized in Table 4.

Question 1 : Does our proposed ChromFound architecture contribute to performance?

We evaluate the individual impact of each key module in our model, including genome-aware tokenization (Section 3.1), the WPSA layer (Section 3.2.1) and the Mamba block (Section 3.2.2). In rows 2-4 of Table 4, we observe a clear performance drop in all clustering metrics compared to row 1, demonstrating that each component contributes complementarily to overall performance.

Question 2 : Does scaling to large-scale data provide performance gains?

As shown in rows 5-6 of Table 4, reducing the data size from 1.97 million to 0.2 million and 20 thousand cells leads to consistent degradation in all evaluation metrics, confirming that large-scale data learning facilitates better generalization.

Question 3 : Is long-context modeling necessary for high-quality representations?

As seen in rows 7-8 of Table 4, reducing the input length by half and by a quarter significantly hurts performance, highlighting the importance of long-context modeling for dynamic OCR landscapes.

Table 4: Ablation study. Red row indicates the full model of ChromFound. The other colored rows correspond to the topics discussed in Section 4.4.

| Genome info | WPSA | Mamba | Data size (million) | # of OCRs per cell | FLOPs(10e9) per cell | ARI(↑) | FMI(↑) | NMI(↑) | AMI(↑) |
|---|---|---|---|---|---|---|---|---|---|
| ✔ | ✔ | ✔ | 1.97 | 440000 | 19.72 | **0.6953±0.0035** | **0.7601±0.0023** | **0.7860±0.0007** | **0.7852±0.0008** |
|  | ✔ | ✔ | 1.97 | 440000 | 19.72 | 0.6452±0.0149 | 0.7095±0.0098 | 0.7227±0.0021 | 0.7301±0.0017 |
|  |  | ✔ | 1.97 | 440000 | 19.72 | 0.5897±0.0086 | 0.6401±0.0113 | 0.6419±0.0012 | 0.6476±0.0015 |
|  | ✔ |  | 1.97 | 440000 | 19.72 | 0.3075±0.0019 | 0.3098±0.0011 | 0.3101±0.0011 | 0.3326±0.0013 |
| ✔ | ✔ | ✔ | 0.2 | 440000 | 19.72 | 0.6539±0.0057 | 0.7261±0.0037 | 0.7593±0.0012 | 0.7583±0.0012 |
| ✔ | ✔ | ✔ | 0.02 | 440000 | 19.72 | 0.6142±0.0053 | 0.6995±0.0034 | 0.7354±0.0015 | 0.7345±0.0015 |
| ✔ | ✔ | ✔ | 1.97 | 220000 | 9.87 | 0.4012±0.0047 | 0.4500±0.0031 | 0.4535±0.0007 | 0.4425±0.0007 |
| ✔ | ✔ | ✔ | 1.97 | 110000 | 4.93 | 0.3276±0.0051 | 0.3589±0.0048 | 0.3674±0.0002 | 0.3668±0.0002 |

## 5 Conclusion

In this work, we present ChromFound, to the best of our knowledge, the first foundation model specifically for scATAC-seq data. To address the inherent challenges of high-dimensionality, sparsity, and dynamic chromatin landscapes, we utilize a hybrid architecture that integrates short- and long-range dependencies with a genome-aware tokenization. Trained on 1.97 million single-cell profiles comprising over 30 human tissue types and 6 disease categories, ChromFound achieves SOTA performance in 6 downstream tasks, showing great robustness in zero-shot cell representation. Its biological significance is further demonstrated by its ability to accurately infer transcriptional responses to enhancer perturbations. In the future, we aim to explore the broader applicability of ChromFound in the large-scale mapping of enhancer-gene regulatory interactions, a critical yet largely uncharted component of the *cis*-regulatory landscape of the human genome.

## 6 Acknowledgments

The authors gratefully acknowledge the following funding sources: AI for Science Program, Shanghai Municipal Commission of Economy and Information, National Natural Science Foundation of China (Grant No. 82394432, 92249302), Shanghai Municipal Science and Technology Major Project (2023SHZDZX02), the King Abdullah University of Science and Technology (KAUST) Office of Research Administration (ORA) under Award No REI/1/5234-01-01, REI/1/5414-01-01, REI/1/5289-01-01, REI/1/5404-01-01, REI/1/5992-01-01, URF/1/4663-01-01, Center of Excellence for Smart Health (KCSH) under award number 5932, and Center of Excellence on Generative AI under award number 5940. The computations in this research were performed using the Computing for the Future at Fudan (CFFF) platform of Fudan University.

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

# A  Discussion of comparing OCR tokenization with other peak-calling algorithms

Traditional scATAC-seq pipelines often rely on a fixed OCR vocabulary (e.g., from a reference atlas) to harmonize datasets, which ensures alignment consistency but introduces two major limitations: (1) a loss of OCR length information, which encodes accessibility strength and resolution, and (2) reduced generalizability, as tissue-specific or novel OCRs are often excluded. ChromFound addresses these issues through a genomic-coordinate–based tokenization strategy that encodes each OCR by its actual genomic start and end positions using sinusoidal positional embeddings (Section 3.1). This position-aware representation naturally accommodates variable-length OCRs and preserves their spatial relationships, enabling the model to capture regulatory heterogeneity across tissues.

We perform an experiment to directly compare ChromFound with VAE-based models trained on reference-peak-aligned data. Specifically, we adopt the cPeaks reference set [45], which defines 1,657,194 peaks across the human genome. We use a 20,000-cell PBMC subset (same as in Table 4, Row 6) as the training data. After mapping peaks to the cPeaks reference and filtering out peaks and cells with all-zero values, we obtain a training set of 306,784 peaks and approximately 18,000 cells.

We train four VAE-based baselines [73, 72, 1, 44] on the aligned data and compare their performance on zero-shot cell clustering (Table 1) against ChromFound trained on the same aligned data. Evaluation metrics and datasets follow Table1, all aligned to 306,784 peaks. Results are summarized in Table 5.

This comparison yields two key observations. First, reference-based alignment substantially limits cross-tissue generalization: only 30% of peaks in retina, 35% in cortex, and 70% in heart overlap with the cPeaks-aligned PBMC training peaks, leading to markedly lower performance of the baseline VAEs on unseen tissues. Second, ChromFound, though robust even when trained on the reference-aligned data, performs best when using its native OCRs, suggesting that fixed-peak harmonization fails to capture novel or shifted OCRs arising from tissue heterogeneity, disease-specific variation, or batch effects.

# B  Experiments details

## B.1  Cell clustering

Since ChromFound is configured in a zero-shot mode for the cell clustering task, no additional fine-tuning architecture or training parameters need to be specified. We directly propagate the data through the encoder structure of ChromFound, extracting the encoder output as the final representation. After performing a pooling layer on the hidden dimension, we apply Principal Component Analysis (PCA) to reduce the dimensionality of the output to 50. Based on this, we compute the metrics for cell clustering and generate the UMAP clustering visualization. The benchmark methods are implemented with the default parameters provided by their source codes.

We compute the inference speed of ChromFound on a node equipped with 32 CPU cores, 256 GB of memory, and one NVIDIA A100 GPU. The input scATAC-seq dataset with 232,354 OCRs. The maximum CPU memory usage is 23.9 GB and the maximum GPU memory consumption is 12.7 GB out of 80 GB available. The inference speed with batch size 4 is 0.97 seconds per batch.

## B.2  Cell type annotation

Building upon our four-layer pretrained model, we extract the encoder output as a general-purpose representation for downstream cell type annotation. To align the model with this task, we incorporate additional fine-tuning layers. Specifically, the hidden dimension produced by ChromFound's encoder is expanded from 128 to 256 before being projected down to a single dimension, resulting in a pooling tensor. Subsequently, additional multilayer perceptron (MLP) layers are employed to predict the logits corresponding to cell types. We freeze the decoder parameters from pretraining while enabling gradient flow through the rest of the pretrained backbone so that it can adapt to the new classification objective. Dropout mitigates overfitting by randomly zeroing hidden units, while LayerNorm helps stabilize and accelerate fine-tuning. The benchmark methods are implemented with the default parameters provided by their source codes.

Table 5: Results of VAE-based models with cPeaks-aligned training against ChromFound. Result style: **best**, second best

| Dataset | Model | VAE | cPeaks | ARI(↑) | FMI(↑) | NMI(↑) | AMI(↑) |
|---|---|---|---|---|---|---|---|
| Cortex(batch 1) | SCALE [73] | ✔ | ✔ | 0.2932 | 0.5050 | 0.4127 | 0.4081 |
| | SCALEX [72] | ✔ | ✔ | 0.4230 | 0.6078 | 0.5543 | 0.5513 |
| | peakVI [1] | ✔ | ✔ | 0.3690 | 0.5643 | 0.5023 | 0.4983 |
| | poissonVI [44] | ✔ | ✔ | 0.3364 | 0.5408 | 0.4636 | 0.4594 |
| | ChromFound | | ✔ | 0.5858 | 0.7244 | 0.7519 | 0.7499 |
| | ChromFound | | | **0.6475** | **0.7503** | **0.7701** | **0.7689** |
| Cortex(batch 2) | SCALE [73] | ✔ | ✔ | 0.2484 | 0.4052 | 0.3657 | 0.3609 |
| | SCALEX [72] | ✔ | ✔ | 0.4481 | 0.5679 | 0.5742 | 0.5710 |
| | peakVI [1] | ✔ | ✔ | 0.3610 | 0.5035 | 0.4876 | 0.4835 |
| | poissonVI [44] | ✔ | ✔ | 0.2820 | 0.4374 | 0.4104 | 0.4058 |
| | ChromFound | | ✔ | 0.5751 | 0.6720 | 0.7145 | 0.7123 |
| | ChromFound | | | **0.6092** | **0.7015** | **0.7276** | **0.7259** |
| Heart(av 3) | SCALE [73] | ✔ | ✔ | 0.3543 | 0.4817 | 0.5228 | 0.5195 |
| | SCALEX [72] | ✔ | ✔ | 0.2795 | 0.4150 | 0.4787 | 0.4752 |
| | peakVI [1] | ✔ | ✔ | 0.3023 | 0.4437 | 0.4816 | 0.4779 |
| | poissonVI [44] | ✔ | ✔ | 0.3778 | 0.5025 | 0.5446 | 0.5414 |
| | ChromFound | | ✔ | 0.5400 | 0.6347 | 0.6588 | 0.6567 |
| | ChromFound | | | **0.5642** | **0.6601** | **0.6981** | **0.6953** |
| Heart(av 10) | SCALE [73] | ✔ | ✔ | 0.4051 | 0.6506 | 0.4461 | 0.4453 |
| | SCALEX [72] | ✔ | ✔ | 0.3013 | 0.5685 | 0.4199 | 0.4191 |
| | peakVI [1] | ✔ | ✔ | 0.3966 | 0.6080 | 0.5018 | 0.5011 |
| | poissonVI [44] | ✔ | ✔ | 0.3939 | 0.5997 | 0.5010 | 0.5003 |
| | ChromFound | | ✔ | 0.6150 | 0.7695 | 0.6968 | 0.6963 |
| | ChromFound | | | **0.6432** | **0.7869** | **0.7220** | **0.7223** |
| Retina(D026_13) | SCALE [73] | ✔ | ✔ | 0.3065 | 0.5771 | 0.4447 | 0.4440 |
| | SCALEX [72] | ✔ | ✔ | 0.2430 | 0.5149 | 0.4127 | 0.4117 |
| | peakVI [1] | ✔ | ✔ | 0.2514 | 0.5260 | 0.4651 | 0.4644 |
| | poissonVI [44] | ✔ | ✔ | 0.2347 | 0.5122 | 0.4063 | 0.4055 |
| | ChromFound | | ✔ | 0.4888 | 0.7081 | 0.7241 | 0.7238 |
| | ChromFound | | | **0.5802** | **0.7549** | **0.7410** | **0.7403** |
| Retina(D19D008) | SCALE [73] | ✔ | ✔ | 0.3143 | 0.5092 | 0.4220 | 0.4210 |
| | SCALEX [72] | ✔ | ✔ | 0.2490 | 0.4542 | 0.3920 | 0.3909 |
| | peakVI [1] | ✔ | ✔ | 0.2664 | 0.4693 | 0.4745 | 0.4736 |
| | poissonVI [44] | ✔ | ✔ | 0.2396 | 0.4473 | 0.3937 | 0.3926 |
| | ChromFound | | ✔ | 0.5368 | 0.6839 | 0.7664 | 0.7660 |
| | ChromFound | | | **0.5956** | **0.7202** | **0.7832** | **0.7868** |
| PBMC(VIB_10xv1_1) | SCALE [73] | ✔ | ✔ | 0.5769 | 0.6756 | 0.7066 | 0.7057 |
| | SCALEX [72] | ✔ | ✔ | 0.5718 | 0.6514 | 0.7205 | 0.7196 |
| | peakVI [1] | ✔ | ✔ | 0.5306 | 0.6249 | 0.7057 | 0.7046 |
| | poissonVI [44] | ✔ | ✔ | 0.5593 | 0.6479 | 0.6742 | 0.6732 |
| | ChromFound | | ✔ | 0.5999 | 0.6827 | 0.7434 | 0.7424 |
| | ChromFound | | | **0.6142** | **0.6995** | **0.7354** | **0.7345** |
| PBMC(BIO_ddseq_1) | SCALE [73] | ✔ | ✔ | 0.4031 | 0.5797 | 0.5380 | 0.5374 |
| | SCALEX [72] | ✔ | ✔ | 0.3326 | 0.5430 | 0.5465 | 0.5458 |
| | peakVI [1] | ✔ | ✔ | 0.3830 | 0.5804 | 0.5737 | 0.5731 |
| | poissonVI [44] | ✔ | ✔ | 0.3959 | 0.5917 | 0.5509 | 0.5503 |
| | ChromFound | | ✔ | 0.4347 | 0.6237 | 0.5867 | 0.5861 |
| | ChromFound | | | **0.4579** | **0.6421** | **0.5902** | **0.5910** |

For experimental settings, we divide the training data into 90% for training and 10% for validation. We train for 20 epochs using the AdamW optimizer with an initial learning rate of $5 \times 10^{-4}$ and a linear warm-up schedule of 50 steps. The best model is chosen based on validation set performance and evaluated on the test set to obtain the reported accuracy and macro F1 score. All experiments are conducted on a single machine equipped with four NVIDIA A100 GPUs.

### B.3 Cross-omics prediction

We employ a two-layer Multilayer Perceptron (MLP) to perform the modality prediction task, with the hidden layer size set to 1024. The learning rate is configured at 1e-4, and AdamW is utilized as the optimizer. A total of 10 epochs are conducted for training. All benchmark methods are implemented with the default parameters in the DANCE package [16]. All experiments are conducted on a single machine equipped with four NVIDIA A100 GPUs. We divide the whole dataset into 80% for training, 10% for evaluation, and 10% for testing.

### B.4 Biological application

To validate the biological relevance of predicted enhancer-gene interactions, we simulate enhancer knockdown using our cross-omics model trained on the Zhu11K dataset [87]. Specifically, we select 141 K562 cells from the test split and iteratively set the accessibility of each candidate enhancer to zero, mimicking CRISPRi-mediated repression. The model then infers post-perturbation gene expression, and the change relative to the unperturbed state is computed and averaged across all cells. The absolute magnitude of this change indicates the likelihood of a regulatory interaction, while the signed change reflects its direction and strength.

We focus on two cancer-related genes, COPZ1 and HNRNPA1, which are extensively characterized in the Fulco4K CRISPRi dataset [87]. Each gene is associated with 117 candidate enhancers, among which 6 (COPZ1) and 10 (HNRNPA1) are experimentally validated. To resolve discrepancies in genomic coordinates between Fulco4K [19] and Zhu11K [87], we perform enhancer mapping using bedtools [53]. In total, the simulation spans 230,819 enhancers and 17,476 genes.

To evaluate performance, we compute the area under the ROC curve seen in Table 4(a) between the absolute expression change and the binary labels of enhancer-gene links tested with a statistical significance on gene expression at a false discovery rate (FDR) < 0.05 [19]. ChromFound achieves the highest accuracy, with AUCs of 0.77 for COPZ1 and 0.61 for HNRNPA1, demonstrating its ability to identify functional regulatory links. Additionally, we calculate the Pearson correlation between signed responses and CRISPRi-measured quantitative effects plotted in Table 4(b).

The Benchmark methods BABEL [71] and CMAE [78] are included for comparison due to their strong cross-omics performance on Zhu11K. However, both rely on filtering the input scATAC-seq profiles to the top 10,000 highly variable OCRs [16], which excludes the majority of candidate enhancers from Fulco4K. As a result, these methods often fail to produce meaningful predictions for enhancer perturbations, yielding near-zero expression changes in most cases. This limitation is evident in both the ROC and scatter plots, underscoring the advantage of ChromFound's genome-wide modeling in capturing fine-grained regulatory effects at scale.

## C Details of datasets

We collect a large-scale human scATAC-seq dataset that includes more than 2.65 million cells and 1.75 trillion tokens. ChromFound for pretraining is based on two fundamental datasets, the human atlas [83] and the fetal atlas [17], which contribute 1.32 million cells spanning multiple human organs and serve as the primary sources for pretraining ChromFound. We accept the CRCh38 as reference genome across all datasets to maintain uniformity, converting datasets originally in hg19 to hg38 when necessary. The resources of all datasets are detailed in Table 6.

## D Pre-processing of scATAC-seq data

We develop a preprocessing pipeline for scATAC-seq data to address sparsity and high dimensionality, ensuring high-quality cells and OCRs for model pretraining and downstream analyses. Preprocessing

is applied after train/test splitting to avoid information leakage, and all datasets are stored in `.h5ad` format with genomic OCR coordinates in the `var` schema.

**Quality Control**: We filter out cells with non-zero counts below 1,000 or above 60,000 and remove OCRs with non-zero counts present in less than 5% of all cell types to preserve heterogeneity among rare populations. For unlabeled datasets, pseudo labels are inferred using TF-IDF followed by LSI for OCR filtering.

**Cell aggregation**: To mitigate sparsity, we aggregate neighboring cells identified through Latent Semantic Indexing (LSI) and cosine similarity. LSI extracts $n$ components after removing technical noise (the first component), and cosine similarity determines the 10 nearest neighbors for aggregation.

**Normalization and Log Transformation**: To stabilize variance and scale the data, we apply total count normalization followed by log transformation.

Table 6: Dataset Information

| Dataset | Tissue | Disease | Cell Numbers | OCR Numbers | Genome | GSE/GSM ID |
|---|---|---|---|---|---|---|
| HumanAtlas [83] | Atlas | Health | 615,998 | 1,154,611 | hg38 | GSE184462 |
| FetalAtlas [17] | Atlas | Health | 707,043 | 1,154,646 | hg38 | GSE149683 |
| PBMCBenchmark [15] | Peripheral Blood | Health | 169,047 | 412,490 | hg38 | GSE194028 |
| Cortex130K [48] | Cortex | Alzheimer's disease | 130,419 | 219,070 | hg38 | GSE174367 |
| Cortex2K [49] | Cortex | Health | 2,174 | 292,156 | hg38 | GSE174226 |
| Zhu45K [86] | Cortex | Health | 45,549 | 304,034 | hg38 | GSE204684 |
| PBMC9K [23] | Peripheral Blood | Leukemia | 9,215 | 108,344 | hg38 | GSE139369 |
| BMMC11K [23] | Bone Marrow | Health | 11,384 | 452,004 | hg19 | GSE194122 |
| Rubin [56] | Epidermis | Health | 288 | 94,633 | hg19 | GSE116428 |
| Xu12K [75] | Breast | Breast cancer | 7,202
5,642 | 87,851
76,850 | hg38 | GSM4798906
GSM4798907 |
| Wang23K [68] | Brain | Glioma | 6,284
129
5,213
5,519
2,229
3,628 | 79,730
1,169
210,434
183,847
40,907
93,121 | hg38 | GSM4119513
GSM4119514
GSM4119515
GSM4119516
GSM4119517
GSM4119518 |
| Buenrostro3K [3] | Bone Marrow | Health | 2,953 | 491,437 | hg19 | GSE96769 |
| Corces0.5K [9] | Bone Marrow/ Peripheral Blood | Leukemia | 576 | 590,650 | hg19 | GSE74310 |
| Corces70K [10] | Brain | Alzheimer's and Parkinson's diseases | 70,631 | 444,747 | hg38 | GSE147672 |
| Satpathy110K [57] | Bone Marrow | Health | 63,882 | 571,400 | hg19 | GSE129785 |
| | Peripheral Blood | Health | 4,146 | 238,616 | | |
| | Peripheral Blood | Health | 4,786 | 152,367 | | |
| | TME | Cancer | 37,818 | 580,789 | | |
| Ziffra77K [88] | Forebrain | Health | 77,354 | 459,953 | hg38 | GSE163018 |
| Kuppe139K [33] | Heart | Myocardial infarction | 139,835 | 429,828 | hg38 | – |
| Liang154K [35] | Retina | Health | 154,775 | 264,833 | hg19 | GSE226108 |
| To326K [65] | Bone | Health | 326,532 | 530,751 | hg38 | – |
| Zhu11K [87] | Cell Line | N/A | 11,632 | 258,044 | hg19 | GSE118912 |
| 10x PBMC Datasets | Peripheral Blood | Health | 10,247
9,688
4,623
1,004
484 | 90,686
144,023
135,377
82,579
65,908 | hg19 | 10k v1.1
10k v2.0
5k v2.0
1k v2.0
500 v2.0 |

# E Supplementary results of ablation study

## E.1 Hyperparameter sensitivity

We evaluate the hyperparameter sensitivity on the cell clustering task using PBMC169K (batch VIB_10xv1_1) dataset.

**Positional embedding temperature**: This parameter controls the frequency of sinusoidal position encodings and can be viewed as a proxy for "genomic resolution". As shown in Table 7, performance slightly drops when temp is within 10e3 to 10e5, and degrades more substantially when temp is too large, likely due to loss of relative position sensitivity.

Table 7: Results of hyperparameter $temp$ sensitivity experiment.

| $temp$ | ARI($\uparrow$) | NMI($\uparrow$) | AMI($\uparrow$) | FMI($\uparrow$) |
|---|---|---|---|---|
| 1000 | 0.6535 | 0.7764 | 0.7755 | 0.7259 |
| 10000 | 0.6512 | 0.7709 | 0.7701 | 0.7243 |
| **100000 (Ours)** | **0.6953** | **0.7860** | **0.7852** | **0.7601** |
| 1000000 | 0.6036 | 0.7344 | 0.7334 | 0.6860 |
| 10000000 | 0.5852 | 0.7269 | 0.7258 | 0.6714 |

**Mamba projection dimension**: This parameter controls the compression within the Mamba block. As shown in Table 8, $D_{low}$ = 32 achieves a strong balance between performance and efficiency. Our choice of 32 is primarily motivated by the trade-off between performance and computational efficiency. Larger values lead to marginal gains but significantly increase FLOPs and memory, with $D_{low}$ = 128 exceeding the memory limits on A100 80G GPUs.

Table 8: Results of Mamba projection dimension $D_{low}$ sensitivity experiment.

| $D_{low}$ | #Parameter | FLOPs (10e11) | Inference Speed (s) | VRAM (GB) | ARI($\uparrow$) |
|---|---|---|---|---|---|
| 16 | 422,593 | 7.41 | 3.4027 | 48.8 | 0.6158 |
| **32 (Ours)** | **450,305** | **7.89** | **3.4624** | **60.7** | **0.6953** |
| 64 | 518,785 | 9.09 | 4.0053 | 72.3 | 0.6927 |
| 128 | 707,969 | 12.4 | OOM | OOM | OOM |

## E.2 Comparison on Transformer-only architectures

Training a vanilla Transformer on scATAC-seq inputs containing nearly one million peaks is practically infeasible due to its quadratic scaling in memory and computation. To approximate a pure Transformer baseline under practical settings, we train two representative single-cell foundation models, Geneformer and scGPT, on the same pretraining corpus with increasing peak lengths (4k–32k) and the same hyperparameters as WPSA. The results of comparison on the cell clustering task using the PBMC169K (batch VIB_10xv1_1) dataset are detailed in Table 9.

Table 9: Results of comparison on Transformer-only architectures.

| Method | OCRs Length | ARI($\uparrow$) |
|---|---|---|
| Geneformer | 4096 | 0.0451 |
|  | 16384 | 0.0457 |
|  | 32768 | 0.0460 |
| scGPT | 4096 | 0.3075 |
|  | 16384 | 0.3774 |
|  | 32768 | 0.3868 |
| **ChromFound** | **440,000** | **0.6953** |

## F  Supplementary results of denoising batch effect

We show the plots of the denoising batch effect as below. The results on To326K [65] from the Bone tissue are plotted in Fig. 5. The results on Kuppel139K [33] from the Heart tissue are plotted in Fig. 6. The results on Morabitol130K [48] from the Cortex tissue are plotted in Fig. 7. The results on PBMC169K [15] from the PBMC tissue are plotted in Fig. 8. In addition, the results of all metrics are listed in Table 10.

Table 10: Comparison of Batch Correction and Biological Conservation Metrics Across Datasets. The best performance for each metric within each dataset is highlighted in bold.

| Dataset | Method | Batch Correction Metrics | | Biological Conservation Metrics | | |
|---|---|---|---|---|---|---|
| | | $ASW_{batch}$ | GraphConn | $ASW_{cell}$ | $ARI_{cell}$ | $NMI_{cell}$ |
| **Cortex** Morabito130K [48] batch 1/2 | scVI [38] | 0.8863 | 0.8595 | 0.5199 | 0.4537 | 0.4631 |
| | PeakVI [1] | 0.9135 | 0.9373 | 0.6022 | 0.8021 | 0.7725 |
| | PoissonVI [44] | 0.9205 | 0.9353 | 0.6021 | 0.7991 | 0.7707 |
| | Scanorama [29] | **0.9391** | 0.9211 | 0.5318 | **0.8234** | 0.7585 |
| | Harmony [32] | 0.9174 | 0.9262 | 0.5793 | 0.8032 | 0.7892 |
| | scANVI [74] | 0.8790 | 0.9697 | **0.6585** | 0.6185 | 0.7916 |
| | Liger [37] | 0.8605 | 0.9598 | 0.5689 | 0.7899 | 0.7870 |
| | SCALEX [72] | 0.9143 | 0.9274 | 0.5791 | 0.7896 | 0.7869 |
| | scBasset [81] | 0.8052 | 0.9784 | 0.6361 | 0.6876 | 0.7230 |
| | **ChromFound** | 0.9323 | **0.9808** | 0.6408 | 0.7531 | **0.8380** |
| **Bone** To326K [65] batch 43/44 | scVI [38] | 0.9073 | 0.9283 | 0.4891 | 0.0070 | 0.0374 |
| | PeakVI [1] | 0.9088 | 0.9259 | 0.5145 | 0.1634 | 0.2167 |
| | PoissonVI [44] | **0.9098** | 0.9259 | 0.5006 | 0.1634 | 0.2167 |
| | Scanorama [29] | 0.8836 | 0.9580 | 0.5115 | 0.2325 | 0.3513 |
| | Harmony [32] | 0.8508 | 0.9544 | 0.5539 | 0.2535 | 0.3066 |
| | scANVI [74] | 0.7939 | 0.9506 | 0.5721 | 0.2835 | 0.4604 |
| | Liger [37] | 0.8244 | 0.8874 | 0.5535 | 0.2167 | 0.3668 |
| | SCALEX [72] | 0.7452 | 0.9506 | 0.5616 | 0.2835 | 0.4604 |
| | scBasset [81] | 0.5163 | 0.7781 | 0.4339 | 0.0603 | 0.1035 |
| | **ChromFound** | 0.8694 | **0.9885** | **0.6027** | **0.6426** | **0.6772** |
| **Heart** Kuppe139K [33] donar av3/av10 | scVI [38] | 0.8128 | 0.8704 | 0.6247 | 0.8186 | 0.8355 |
| | PeakVI [1] | 0.9149 | 0.5980 | 0.5215 | 0.6076 | 0.5680 |
| | PoissonVI [44] | **0.9460** | 0.5980 | 0.5106 | 0.6076 | 0.5680 |
| | Scanorama [29] | 0.9378 | 0.7804 | 0.4707 | 0.6869 | 0.7072 |
| | Harmony [32] | 0.8490 | 0.8407 | 0.5546 | 0.8066 | 0.8298 |
| | scANVI [74] | 0.7844 | 0.7715 | 0.4337 | 0.5070 | 0.6553 |
| | Liger [37] | 0.8374 | 0.8407 | 0.4196 | 0.4875 | 0.6287 |
| | SCALEX [72] | 0.8638 | 0.8164 | 0.5424 | 0.7296 | 0.7711 |
| | scBasset [81] | 0.7756 | 0.8282 | **0.6387** | 0.8055 | 0.7417 |
| | **ChromFound** | 0.8589 | **0.8770** | 0.6155 | **0.9406** | **0.8980** |
| **PBMC** 169K [15] HAR ddseq 1/2 | scVI [38] | 0.7452 | 0.8557 | 0.5794 | 0.5286 | 0.6888 |
| | PeakVI [1] | 0.7511 | 0.8266 | 0.6054 | 0.5456 | 0.7005 |
| | PoissonVI [44] | 0.7895 | 0.8266 | 0.5626 | 0.5456 | 0.7005 |
| | Scanorama [29] | 0.7964 | 0.8066 | 0.5613 | 0.5456 | 0.7005 |
| | Harmony [32] | 0.7745 | 0.8120 | 0.5711 | 0.5525 | **0.7069** |
| | scANVI [74] | 0.7480 | 0.8448 | 0.6223 | 0.5114 | 0.6773 |
| | Liger [37] | 0.7755 | 0.8091 | 0.6197 | 0.5481 | 0.7020 |
| | SCALEX [72] | **0.8221** | 0.7850 | 0.5534 | **0.5549** | 0.7003 |
| | scBasset [81] | 0.7753 | 0.8266 | 0.5695 | 0.5456 | 0.7005 |
| | **ChromFound** | 0.7923 | **0.8511** | **0.6443** | 0.5730 | 0.7156 |

## G  Analysis of Cell Type Annotation

### G.1  Metric Value of Cell Type Annotation

To adapt ChromFound for cell type annotation, we leverage its four-layer pretrained model, extracting the encoder output as a general-purpose representation. The encoder's hidden dimension is expanded

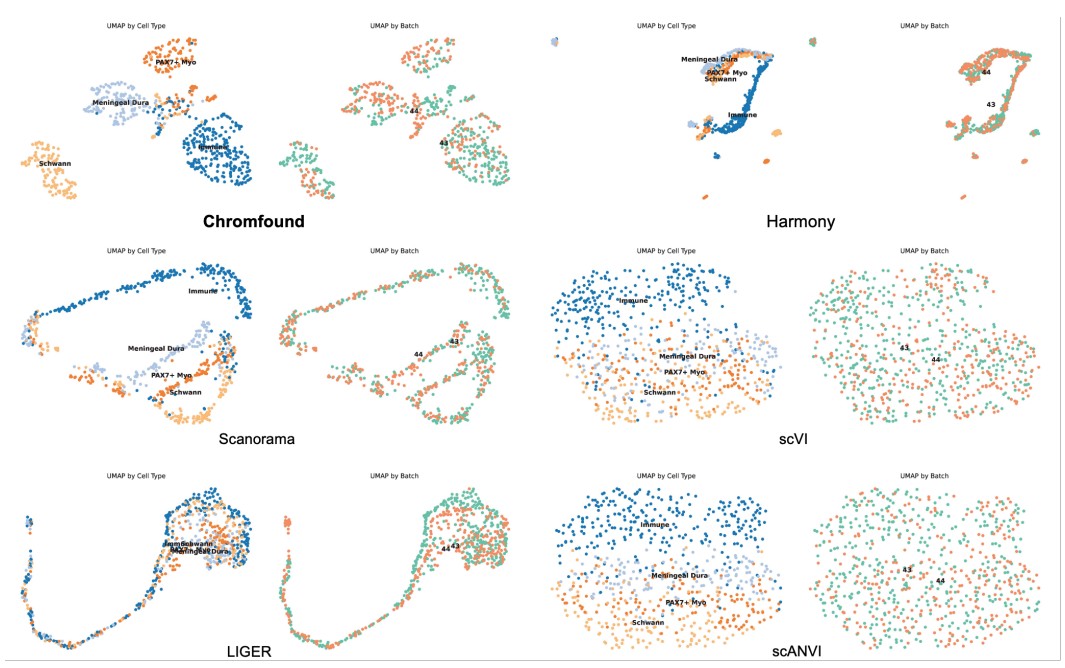

Figure 5: The plots illustrate ChromFound's superior performance in denoising batch effect on **Bone** To326K [65].

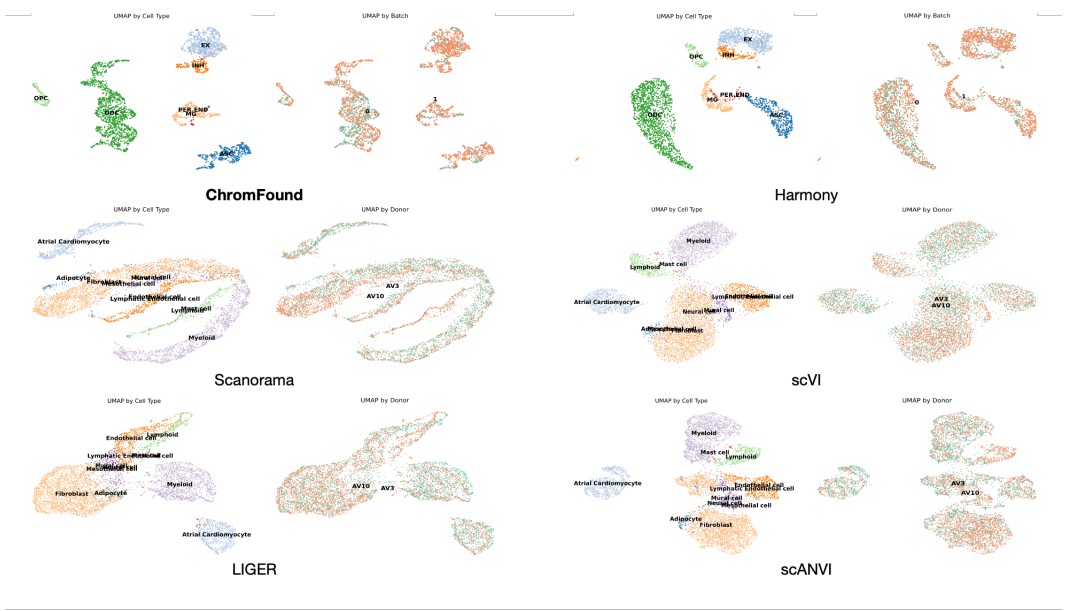

Figure 6: The plots illustrate ChromFound's superior performance in denoising batch effect on **Heart** Kuppel139K [33].

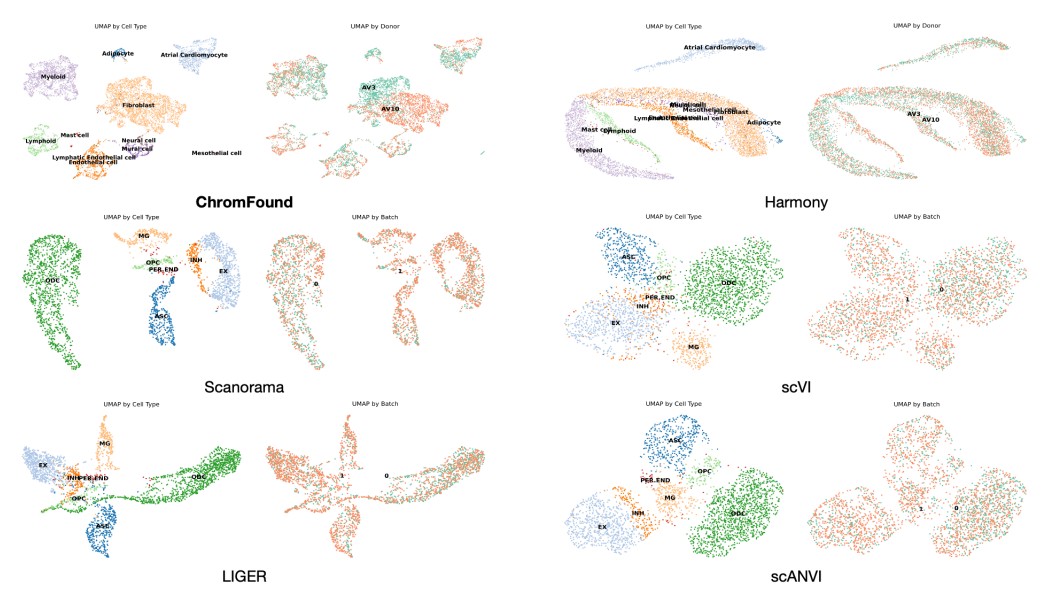

Figure 7: The plots illustrate ChromFound's superior performance in denoising batch effect on **Cortex** Morabitol130K [48].

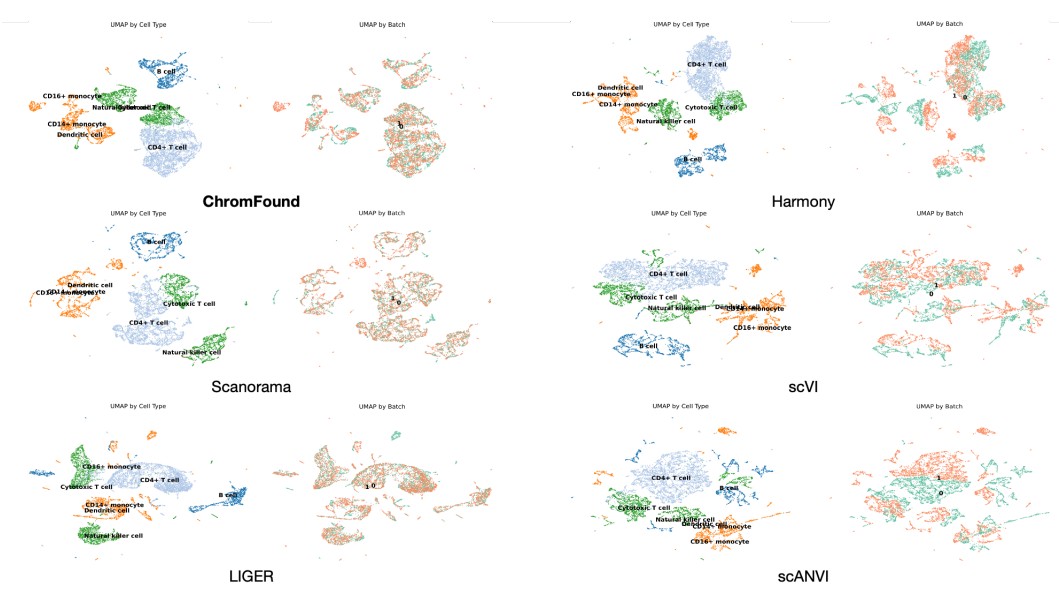

Figure 8: The plots illustrate ChromFound's superior performance in denoising batch effect on **PBMC** 169K [15].

from 128 to 256 and projected to a single dimension, forming a pooling tensor. This tensor is then processed by additional multilayer perceptron (MLP) layers to predict cell type logits. During fine-tuning, the pretrained decoder parameters are frozen, while gradients flow through the rest of the backbone to align with the classification objective. Dropout and LayerNorm are applied to mitigate overfitting and stabilize training, respectively. Benchmark methods are implemented using their default parameters as provided in their source codes.

For the experimental setup, we split the training data into 90% for training and 10% for validation. Training is conducted over 20 epochs using the AdamW optimizer with an initial learning rate of $5 \times 10^{-4}$ and a 50-step learning rate warmup schedule. The model is selected based on validation performance and evaluated on the test set to compute accuracy and macro F1-score. For certain configurations, such as training on EPF_hydrop_2 and testing on VIB_10xv1_1, the learning rate is reduced to $2.5 \times 10^{-4}$ to ensure stable convergence. All experiments are performed on a single machine with four NVIDIA A100 GPUs.

### G.2   Confusion Matrix of Cell Type Annotation

In the cell classification task where EPF_hydrop_3 served as the training set and VIB_10xv1_2 as the test set, ChromFound achieves a 4.71% improvement in accuracy and a 14.69% increase in macro F1 score compared to the previous SOTA method, CellCano. To further elucidate the specific advancements of ChromFound, we present the confusion matrices of both methods for a more detailed comparative analysis.

Here are some biological insights from the comparison of confusion matrices:

(1) The observed misclassification rate of natural killer cells as cytotoxic T cells by the Cellcano model (19.35%) and its significant reduction by ChromFound (2.22%) highlights the importance of accurately distinguishing these functionally related yet distinct immune cell populations. Natural killer cells and cytotoxic T cells share cytotoxic properties, as both can mediate target cell lysis through perforin and granzyme pathways. However, they differ fundamentally in their ontogeny, activation mechanisms, and immune regulation.

(2) The high misclassification rate of CD16+ monocytes as CD14+ T cells by Cellcano (91.91%), and its substantial reduction by ChromFound (66%), underscores the challenge of accurately distinguishing myeloid from lymphoid lineages in cell type annotation based on scATAC-seq. CD16+ monocytes, a subset of non-classical monocytes, exhibit distinct chromatin accessibility patterns associated with Fc receptor signaling, inflammatory responses, and patrolling behavior, whereas CD14+ T cells, a less well-characterized subset, retain a predominantly lymphoid epigenetic signature. The improved classification by ChromFound suggests a refined ability to resolve lineage-specific regulatory elements, likely by better capturing differential enhancer accessibility and transcription factor binding landscapes unique to myeloid versus lymphoid cell fate.

(3) Cellcano misclassifies dendritic cells as CD14+ monocytes at a rate of 65.7% while ChromFound reduces this misclassification to 53.33%. The misclassification of dendritic cells as CD14+ monocytes is particularly interesting because both cell types share similar transcriptional signatures, especially in immune responses. dendritic cells and monocytes both play critical roles in antigen presentation and inflammation, which may lead to similarities in the chromatin accessibility profiles captured by scATAC-seq. From a biological perspective, this reduction in misclassification also underscores the importance of distinguishing between functionally distinct but phenotypically similar immune cell subsets. Dendritic cells, being key mediators of immune tolerance and initiation, have distinct regulatory networks compared to monocytes, which are more directly involved in inflammatory responses. By refining the accuracy of cell type annotation, ChromFound enables a more precise understanding of immune cell dynamics, particularly in the context of immune responses and disease progression.

## H   Benchmark Methods for Downstream Tasks

This section describes the benchmark methods used for evaluating the performance of ChromFound across various downstream tasks. These methods are selected based on their established effectiveness in single-cell data analysis, spanning cell clustering, cell type annotation, and cross-omics modality prediction.

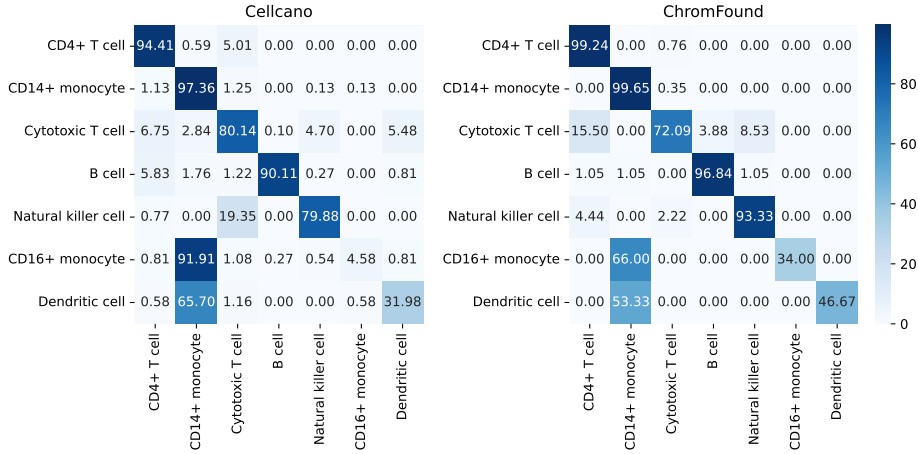

Figure 9: Confusion matrix for cell type annotation task with a comparison of Cellcano and Chrom-Found.

## H.1 Cell Clustering

Cell clustering methods aim to group single-cell data into meaningful clusters, often leveraging deep learning and generative models to handle the high dimensionality and sparsity of scATAC-seq data.

### H.1.1 SCALE

SCALE [73] is a deep generative framework that employs probabilistic Gaussian Mixture Models to analyze high-dimensional scATAC-seq data. It outperforms existing tools in clustering by effectively capturing latent structures in the data.

### H.1.2 SCALEX

SCALEX [72] is a deep learning method designed for integrating single-cell data. It projects cells into a batch-invariant embedding space in an online manner, enabling robust cell representation and achieving excellent clustering performance.

### H.1.3 CASTLE

CASTLE [12] is a deep generative model tailored for single-cell epigenomic data. It uses a vector-quantized variational autoencoder framework [66] to extract discrete latent embeddings, excelling in cell clustering tasks by addressing the challenges of high dimensionality and sparsity.

### H.1.4 scBasset

scBasset [81] is a sequence-based convolutional neural network method designed for modeling single-cell ATAC-seq data. By utilizing DNA sequence information from accessibility peaks and the expressive power of neural networks, scBasset achieves superior performance in tasks such as cell type identification and data integration across single-cell ATAC-seq and multiome datasets.

### H.1.5 PeakVI

PeakVI [1] is a variational autoencoder (VAE)–based probabilistic model for single-cell chromatin accessibility data. It captures both biological and technical variation through a hierarchical generative process, enabling effective batch correction and denoising. The learned latent representation supports diverse downstream tasks such as clustering and differential accessibility analysis, providing a strong probabilistic baseline for scATAC-seq modeling.

### H.1.6 PoissonVI

PoissonVI [44] extends the VAE framework by modeling raw accessibility counts with a Poisson likelihood instead of binarized inputs. This design preserves quantitative accessibility information and improves sensitivity to subtle regulatory differences. PoissonVI achieves robust performance across clustering and integration tasks while maintaining a clear probabilistic interpretation.

### H.1.7 SnapATAC2

SnapATAC2 [84] is a scalable framework for large-scale scATAC-seq analysis. It represents each cell by a high-dimensional accessibility profile and applies diffusion- and spectral-based dimensionality reduction to reveal chromatin structure. With optimized memory management, parallel I/O, and GPU acceleration, SnapATAC2 efficiently handles datasets with millions of cells and serves as a strong non–deep learning baseline.

### H.1.8 Signac

Signac [59] is an R package built on the Seurat ecosystem for single-cell chromatin accessibility analysis. It integrates preprocessing, feature selection, dimensionality reduction, and visualization within a unified workflow, and supports multimodal integration with scRNA-seq data. Owing to its usability and reproducibility, Signac remains a widely adopted toolkit for exploratory and integrative single-cell epigenomic analysis.

## H.2 Denoising Batch Effect

Batch effect removal methods aim to align single-cell data across different batches while preserving biological signals, addressing technical variations that can obscure true biological differences. In this evaluation, we also benchmark SCALEX [72], previously introduced for cell clustering, and scBasset [81], noted for its sequence-based modeling, alongside other methods. These are included in the comparison for batch effect correction within their respective frameworks, with their detailed definitions provided in prior sections.

### H.2.1 Harmony

Harmony [32] is a prominent batch correction technique in single-cell analysis, designed to harmonize datasets from diverse experimental batches. It employs an iterative strategy to align cells into a shared low-dimensional embedding, optimizing the similarity of cell types across batches while retaining biological variability, making it highly effective for downstream analyses in single-cell genomics.

### H.2.2 scVI

scVI [38] is a package for end-to-end analysis of single-cell omics data. The package is composed of several deep generative models for omics data analysis.

### H.2.3 Scanorama

Scanorama [29] is designed to be used in scRNA-seq pipelines downstream of noise-reduction methods, including those for imputation and highly-variable gene filtering. The results of Scanorama integration and batch correction can then be used as input to other tools for clustering, visualization, and analysis of scRNA sequences.

### H.2.4 scANVI

scANVI [74] (single-cell ANnotation using Variational Inference; Python class SCANVI) is a semi-supervised model for single-cell transcriptomics data. In a sense, it can be seen as a scVI [38] extension that can leverage the cell type knowledge for a subset of the cells present in the data sets to infer the states of the rest of the cells. For this reason, scANVI can help annotate a data set of unlabelled cells from manually annotated atlases.

### H.2.5 Liger

Liger [37] (Linked Inference of Genomic Experimental Relationships) integrates multi-omics single-cell data via integrative non-negative matrix factorization (iNMF) to define shared cell identities across protocols/species.

## H.3 Cell Type Annotation

Cell type annotation methods focus on supervised learning approaches to assign cells to predefined classes, leveraging advanced neural network architectures for improved accuracy.

### H.3.1 Cellcano

Cellcano [42] utilizes gene expression scores as input and employs a dual-phase training framework to achieve enhanced accuracy in cell type annotation across diverse datasets.

### H.3.2 EpiAnno

EpiAnno [7] implements a probabilistic generative model with a Bayesian neural network. It delivers remarkable performance in cell type annotation, particularly when applied to diverse single-cell datasets.

## H.4 Cross-omics Prediction

Cross-omics modality prediction methods aim to infer one modality (e.g., ATAC-seq) from another (e.g., RNA-seq), often using integrative neural network frameworks to model the relationships between modalities.

### H.4.1 BABEL

BABEL [71] leverages an interoperable neural network model to translate between the transcriptome and chromatin profiles of individual cells, enabling effective cross-omics prediction.

### H.4.2 CMAE

CMAE [78] learns a probabilistic coupling between different data modalities using autoencoders. It provides a robust framework for integrating and translating between single-cell data modalities.

### H.4.3 scMoGNN

scMoGNN [69], an official winner in the overall ranking of modality prediction from the NeurIPS 2021 Competition, presents a general Graph Neural Network framework to facilitate multimodal single-cell data analysis, demonstrating superior performance in cross-omics prediction tasks.

### H.4.4 scMM

scMM [47] is a novel deep generative model-based framework for single-cell multi-omics data analysis (e.g., transcriptome and chromatin accessibility). It employs a mixture-of-experts multimodal approach to extract interpretable joint representations and enable cross-modal generation, achieving end-to-end learning by modeling raw count data with modality-specific probability distributions.

## H.5 Biological Application: Predicting Enhancer-Gene Link and Perturbation Response

This section outlines the benchmark methods evaluated for the biological application of predicting enhancer-gene links and perturbation responses using ChromFound. This task leverages single-cell data to infer regulatory relationships and responses to perturbations, with a focus on maintaining biological relevance. BABEL [71] and CMAE [78], the second best models in cross-omics prediction tasks, are included in this application aim to model the regulatory interactions between enhancers and genes,

# I  Evaluation Metrics for Downstream Tasks

This appendix details the evaluation metrics used for the downstream tasks in our study, organized by task. Each metric is defined mathematically to ensure clarity and reproducibility, with ranges and performance interpretations provided where applicable.

## I.1  Cell Clustering

Cell clustering is an unsupervised learning task that groups single-cell data into clusters. When ground-truth labels (e.g., cell types) are available, we evaluate clustering performance using the following metrics.

### I.1.1  Adjusted Rand Index (ARI)

The Rand Index (RI) measures the proportion of correctly grouped sample pairs:

$$\text{RI} = \frac{\text{TP} + \text{TN}}{\text{TP} + \text{TN} + \text{FP} + \text{FN}},$$

where TP, TN, FP, and FN denote the numbers of true positives, true negatives, false positives, and false negatives, respectively, based on whether pairs are correctly grouped together or apart in both the clustering and ground-truth labels. The Adjusted Rand Index (ARI) corrects for chance by adjusting the expected RI under random labeling:

$$\text{ARI} = \frac{\text{RI} - \mathbb{E}[\text{RI}]}{\max(\text{RI}) - \mathbb{E}[\text{RI}]},$$

ranging from $-1$ to $1$, where higher values indicate better agreement between clustering results and ground-truth labels.

### I.1.2  Fowlkes-Mallows Index (FMI)

The Fowlkes-Mallows Index (FMI) combines precision and recall for clustering:

$$\text{FMI} = \sqrt{\frac{\text{TP}}{\text{TP} + \text{FP}} \cdot \frac{\text{TP}}{\text{TP} + \text{FN}}},$$

where TP, FP, and FN are defined as above. It ranges from $0$ to $1$, with higher values indicating better clustering performance.

### I.1.3  Normalized Mutual Information (NMI)

NMI quantifies the shared information between clustering $C$ and ground-truth $Y$:

$$\text{NMI}(Y, C) = \frac{2 \cdot I(Y; C)}{H(Y) + H(C)},$$

where $I(\cdot; \cdot)$ is mutual information and $H(\cdot)$ is entropy. It ranges from $0$ to $1$, with higher values indicating greater alignment between the clustering and ground-truth labels.

### I.1.4  Adjusted Mutual Information (AMI)

AMI adjusts NMI for chance by accounting for expected mutual information under random labeling:

$$\text{AMI} = \frac{I(Y; C) - \mathbb{E}[I(Y; C)]}{\max(I(Y; C)) - \mathbb{E}[I(Y; C)]},$$

ranging from $0$ to $1$, where higher values indicate a more robust agreement between clustering and ground-truth, especially in datasets with many clusters.

## I.2  Denoising Batch Effect

Denoising Batch Effect aligns single-cell data across batches while preserving biological signals. We evaluate performance using biological conservation and batch mixing metrics implemented in [41].

### I.2.1 Biological Conservation Metrics

We use the following metrics to assess the preservation of biological signals, where $C$ denotes the set of cell types.

**Normalized Mutual Information (NMI$_{\text{cell}}$)**

$$\text{NMI}_{\text{cell}}(Y, C) = \frac{2 \cdot I(Y; C)}{H(Y) + H(C)},$$

where $I(\cdot; \cdot)$ is mutual information and $H(\cdot)$ is entropy. It ranges from 0 to 1, with higher values indicating better alignment with ground-truth cell types.

**Adjusted Rand Index (ARI$_{\text{cell}}$)**

$$\text{ARI}_{\text{cell}} = \frac{\text{RI} - \mathbb{E}[\text{RI}]}{\max(\text{RI}) - \mathbb{E}[\text{RI}]},$$

where $\text{RI} = \frac{\text{TP+TN}}{\text{TP+TN+FP+FN}}$ and TP, TN, FP, FN are defined as in the clustering section. It ranges from $-1$ to 1, with higher values indicating better clustering agreement with cell types.

**Average Silhouette Width (ASW$_{\text{cell}}$)**

$$\text{ASW}_{\text{cell}} = \frac{1}{N} \sum_{i=1}^{N} \frac{b(i) - a(i)}{\max(a(i), b(i))},$$

where $a(i)$ is the average distance of cell $i$ to others in its cell type, and $b(i)$ is the smallest average distance to another cell type. It ranges from $-1$ to 1, with higher values indicating better preservation of biological clustering.

**Average Biological Score (AvgBIO)**

$$\text{AvgBIO} = \frac{\text{ARI}_{\text{cell}} + \text{NMI}_{\text{cell}} + \text{ASW}_{\text{cell}}}{3},$$

averaging the biological conservation metrics, ranging from 0 to 1, with higher values indicating better preservation of biological signals.

### I.2.2 Batch Mixing Metrics

We assess batch mixing using the following metrics, where $B$ denotes the set of batches.

**Inverse Average Silhouette Width (ASW$_{\text{batch}}$)**

$$\text{ASW}_{\text{batch}} = 1 - \left| \frac{1}{N} \sum_{i=1}^{N} \frac{b(i) - a(i)}{\max(a(i), b(i))} \right|,$$

where $a(i)$ and $b(i)$ are computed with respect to batch labels, with $a(i)$ as the average distance to other cells in the same batch and $b(i)$ as the smallest average distance to cells in another batch. It ranges from 0 to 1, with higher values indicating better batch mixing.

**Graph Connectivity (GraphConn)**

$$\text{GraphConn} = \frac{1}{|C|} \sum_{c \in C} \frac{|\text{LCC}(G_c^{\text{kNN}})|}{N_c},$$

where $\text{LCC}(G_c^{\text{kNN}})$ is the size of the largest connected component in the k-nearest neighbors (kNN) graph of cells in cell type $c$, and $N_c$ is the number of cells in $c$. It ranges from 0 to 1, with higher values indicating better connectivity across batches.

**Average Batch Score (AvgBATCH)**

$$\text{AvgBATCH} = \frac{\text{ASW}_{\text{batch}} + \text{GraphConn}}{2},$$

averaging the batch mixing metrics, ranging from 0 to 1, with higher values indicating better batch correction.

## I.3   Cell Type Annotation

Cell type annotation is a supervised task that assigns cells to ground-truth cell type. We use the following metrics to evaluate performance.

### I.3.1   Accuracy

Accuracy measures the proportion of correctly classified cells:

$$\text{Accuracy} = \frac{\sum_{c \in C} \text{tp}_c}{\sum_{c \in C} N_c},$$

where $\text{tp}_c$ is the number of true positives for cell type $c$, and $N_c$ is the total number of cells in $c$. It ranges from 0 to 1, with higher values indicating better classification performance.

### I.3.2   Macro F1 Score

The Macro F1 Score averages the F1 scores across all cell types:

$$\text{Macro F1} = \frac{1}{|C|} \sum_{c \in C} F1_c, \quad F1_c = \frac{2 \cdot \text{Precision}_c \cdot \text{Recall}_c}{\text{Precision}_c + \text{Recall}_c},$$

where $\text{Precision}_c = \frac{\text{tp}_c}{\text{tp}_c + \text{fp}_c}$ and $\text{Recall}_c = \frac{\text{tp}_c}{\text{tp}_c + \text{fn}_c}$, with $\text{fp}_c$ and $\text{fn}_c$ as false positives and false negatives for cell type $c$. It ranges from 0 to 1, with higher values indicating better balanced performance across classes.

## I.4   Cross-Omics Prediction

Cross-omics prediction infers one modality (e.g., ATAC-seq) from another (e.g., RNA-seq). We evaluate performance using correlation-based metrics.

### I.4.1   Pearson Correlation Coefficient (PCC)

PCC measures the linear correlation between predicted ($x_i$) and true ($y_i$) values:

$$\text{PCC} = \frac{\sum_{i=1}^{N}(x_i - \bar{x})(y_i - \bar{y})}{\sqrt{\sum_{i=1}^{N}(x_i - \bar{x})^2} \cdot \sqrt{\sum_{i=1}^{N}(y_i - \bar{y})^2}},$$

where $\bar{x}$ and $\bar{y}$ are the means of the predicted and true values, respectively. It ranges from $-1$ to 1, with higher (closer to 1) values indicating stronger positive linear correlation.

### I.4.2   Concordance Correlation Coefficient (CCC)

CCC extends PCC by accounting for bias and scale differences:

$$\text{CCC} = \frac{2 \cdot \rho \cdot \sigma_x \cdot \sigma_y}{\sigma_x^2 + \sigma_y^2 + (\mu_x - \mu_y)^2},$$

where $\rho$ is the PCC, $\sigma_x$ and $\sigma_y$ are the standard deviations of the predicted and true values, and $\mu_x$ and $\mu_y$ are their means. It ranges from $-1$ to 1, with higher (closer to 1) values indicating better agreement in both correlation and scale.

## I.5   Predicting Enhancer-Gene Links and Perturbation Response

These tasks involve predicting regulatory relationships or perturbation outcomes, often treated as classification problems. We use the following metrics.

### I.5.1 Area Under the Receiver Operating Characteristic Curve (AUC-ROC)

The Receiver Operating Characteristic (ROC) curve plots the True Positive Rate (TPR) against the False Positive Rate (FPR) at various classification thresholds:

$$\text{TPR} = \frac{\text{TP}}{\text{TP} + \text{FN}}, \quad \text{FPR} = \frac{\text{FP}}{\text{FP} + \text{TN}},$$

where TP, TN, FP, and FN are true positives, true negatives, false positives, and false negatives, respectively. The Area Under the ROC Curve (AUC-ROC) quantifies the overall performance across all thresholds, ranging from $0$ to $1$, with higher values indicating better classification performance, where $1$ represents perfect classification and $0.5$ represents random guessing.

