# OpenReview forum: "ChromFound: Towards A Universal Foundation Model for Single-Cell Chromatin Accessibiltiy Data"
_NeurIPS.cc/2025/Conference — NeurIPS 2025 poster_

### Official Review · Reviewer_VAFg · 2025-06-29

**Clarity:** 3
**Significance:** 2
**Originality:** 3
**Rating:** 4
**Confidence:** 4

**Summary:**

The authors introduce a foundation model specifically designed for single-cell chromatin accessibility (scATAC-seq) data. The authors propose a hybrid encoder architecture that integrates Mamba blocks with local windowed self-attention and introduces a genome-aware tokenization scheme for encoding open chromatin regions (OCRs). Trained on 1.97 million cells from over 30 tissues and 6 disease conditions, ChromFound is evaluated on a broad set of downstream tasks: zero-shot cell representation, cell type annotation, cross-omics prediction, and enhancer-gene link inference.

**Questions:**

1. The authors should compare Chromfound with linear regression in cell type prediction and cross-omics prediction
2. How do the authors unify the peaks to adapt Chromfound for different datasets?
3. Does Chromfound achieve different performance across different tissues for all tasks?
4. Do the authors compare the performance and computational complexity of WPSA and linear attention?
4. Statistical Significance: Did you do any random seed runs or significance tests for the modest performance gain?

**Ethical Concerns:**

["NO or VERY MINOR ethics concerns only"]

**Final Justification:**

The authors have addressed all my concerns.

**Limitations:**

1. The novelty of model architecture is limited. There is no well-designed module for scATAC-seq data or tabular data, except for the position embeddings and mamba adoption.
2. The training loss is not really suitable for the scATAC-seq data since the contextual signals of other regions are not significant with the dropout events.
3. The downstream tasks are not comprehensive. The authors should conduct more research on perturbation prediction and temporal scATAC-seq prediction.
4. More ablation studies should be conducted.

**Paper Formatting Concerns:**

There is no formatting issue.

**Quality:**

3

**Strengths And Weaknesses:**

Strengths: $\bullet$ This is a pioneer foundation model explicitly for scATAC-seq. $\bullet$ In order to address the high dimension issue of scATAC, the authors propose a novel architectural design combining Mamba and windowed attention. $\bullet$ It demonstrates strong performance across multiple tasks (representation, annotation, prediction).

Weakness:
$\bullet$ The novelty of the model architecture is limited. The difference between Chromefound and other single-cell foundation models is that it adopts Mamba to process long sequential data. $\bullet$ The training loss is not really suitable for the scATAC-seq data since the contextual signals of other regions are not significant with the dropout events. scATAC data is more sparse than scRNA data, however, the foundation model on scRNA does not really achieve robust performance when comparing to single linear regression. $\bullet$ Unlike scRNA data which the gene set is already defined, scATAC fragment sets are not always the same among different datasets by different depths. The authors do not claim how they unify the formats of different datasets to employ Chromfound. $\bullet$ As far as I know, there is another scATAC foundation model Epifoundation(https://www.biorxiv.org/content/10.1101/2025.02.05.636688v1) before Chromefound. $\bullet$  Missing reference like Epifoundation, ATAC-Diff (https://arxiv.org/abs/2408.14801), and GET (https://www.nature.com/articles/s41586-024-08391-z). $\bullet$ In the ablation study, the performance of Chromfound is close with much less training set. I am wondering if less OCR would decrease the performance.

---

> ### Author Rebuttal · Authors · 2025-07-31
>
> We sincerely thank the reviewer for their constructive feedbacks and recognizing our work’s novelty and robust performance across downstream tasks. We would like to clarify and elaborate on the following points of concern:
>
> ## [W1,L1] Model Novelty
> As our submission is under the _Machine Learning for Sciences_ area, our goal is to tailor the architecture to scATAC-seq characteristics for biologically meaningful representation and practical applications. We clarify the design of ChromFound’s each component as follows:
> - Unified OCR tokenization (Sec 3.1) encodes chromosomal identity, genomic coordinates, and continuous-valued accessibility, providing a generalizable architecture for large-scale pretraining of heterogeneous scATAC-seq datasets.
> - WPSA (Sec 3.2.1) captures local _cis_-regulatory dependencies within ±200 kb of transcriptional start sites, aligning with biological enhancer-promoter interactions.
> - Mamba (Sec 3.2.2) complements WPSA by modeling long-range interactions across hundreds of thousands of OCRs per cell, enabling ChromFound to learn both local and global chromatin accessibility patterns.
> - Pretrain objective (Sec 3.3) reconstructs masked both zero and non-zero OCRs using MSE loss, which ensures robustness to the high sparsity of scATAC-seq data.
>
> ## [Q1] Shallow Model Baseline Comparison
>
> We thank the reviewer for the helpful suggestion of including shallow baselines for both cell type annotation and cross-omics prediction.
>
> 1. Cell Type Annotation
>
>     We implement two logistic regression models (sklearn.linear_model.LogisticRegression) using full or HVG-filtered OCRs (5,000 peaks), trained with default hyperparameters. These baselines are evaluated across the same datasets in Fig. 3 and compared against ChromFound. Results are reported in the format of Accuracy/F1 score as below:
>
> |Tissue/train/test|Linear|Linear(HVG5k)|ChromFound|
> |-|-|-|-|
> |PBMC/EPF_hydrop_1/VIB_10xv1_1|0.8132/0.6741|0.4527/0.3189|0.8863/0.8135|
> |PBMC/EPF_hydrop_3/VIB_10xv1_2|0.8227/0.6709|0.4791/0.3355|0.9003/0.8134|
> |Bone/Batch27/Batch26|0.7198/0.6241|0.3884/0.3150|0.8230/0.5477|
> |Bone/Batch43/Batch44|0.7854/0.7873|0.4618/0.4414|0.8368/0.8335|
> |Cortex/Batch2/Batch1|0.8367/0.6031|0.4880/0.2403|0.9366/0.7715|
> |Cortex/Batch3/Batch2|0.8455/0.6598|0.4295/0.2804|0.9188/0.8063|
> |Retina/D19D003/D018_13|0.9217/0.8781|0.7737/0.5235|0.9792/0.9780|
> |Retina/D021_13/D19D003|0.9278/0.8862|0.8542/0.7473|0.9786/0.9762|
>
> 2. Cross-Omics Prediction
>
>     For cross-omics prediction, we implement a shallow two-layer MLP that maps OCR inputs to genome-wide gene expression profiles. Given the complexity of predicting over 30,000 gene targets, we used a hidden layer size of 4096, ReLU activation, learning rate of 1e-4, and 50 training epochs. Results are reported in the format of PCC/CCC as below:
>
> |Model|Cortex_Zhu45K|Bone_To326K|BMMC_multiome_2021|BMMC_atac2gex_2022|Cell_lines_Zhu11K|
> |-|-|-|-|-|-|
> |MLP|0.1022/0.0721|0.0946/0.0709|0.1009/0.0627|0.1023/0.0558|0.2333/0.1729|
> |ChromFound|0.8316/0.8064|0.8304/0.7472|0.4249/0.4032|0.9293/0.8818|0.9449/0.9071|
>
> Key Conclusions:
> 1. ChromFound consistently outperforms logistic regression, a strong baseline for annotation.
> 2. Full OCR input significantly outperforms HVG-filtered input, reinforcing ChromFound's necessity of modeling genome-wide regulatory elements.
> 3. The MLP baseline performs poorly on cross-omics prediction, revealing the limitations of simple linear or shallow models in capturing the complex _cis_-regulatory logic underlying chromatin-to-expression mapping. ChromFound is specifically designed to model this complexity through deep representation learning and large-scale self-supervised pretraining.
>
> ## [W2,L2] Training Loss Suitability
>
> We appreciate the reviewer’s concern regarding sparsity in scATAC-seq data.
> Actually, we have mentioned that ChromFound selectively reconstructs both non-zero and zero OCRs (Sec. 3.3). The masking strategy ensures the training procedure is not affected by the dropout events.
> Followed by xTrimoGene, we use the MSE loss for reconstructing the masking peaks, which is suitable for continuous-valued accessibility data.
> Fig. 2 further demonstrates that ChromFound maintains stable performance under severe downsampling, showcasing its robustness to sparsity.
>
> ## [W3,Q2] OCR Format Unification Across Datasets
>
> Thank you for the question. ChromFound adopts a coordinate-based tokenization strategy (Sec. 3.1), where each OCR is represented by its chromosome ID, start/end positions on GRCh38, and continuous accessibility. This enables ChromFound to unify peaks across datasets without requiring pre-aligned peak sets. Notably, the cross-tissue OCR overlap is quite low, highlighting the necessity of position-aware encoding for robust generalization.
>
> ## [W4,W5] References to Related Works
>
> We sincerely thank the reviewer for pointing out these three important related works: Epifoundation, ATAC-Diff, and GET.
> - Epifoundation is quite impressive work, building a foundation model by leveraging paired scATAC-seq and scRNA-seq data to learn peak-gene alignments. However, its reliance on paired multi-omics data limits scalability as the paired data remains limited.
> In contrast, ChromFound adopts a self-supervised approach that does not require paired modalities, enabling pretraining on over 2.6 million cells from diverse tissues and disease conditions. Furthermore, ChromFound extends beyond Epifoundation’s scope by supporting biological applications such as enhancer-gene link inference and perturbation response prediction.
> - ATAC-Diff is designed for identifying differential chromatin accessibility across conditions instead of supporting zero-shot transfer learning.
> - GET, as we have already acknowledged in Section 2, Line 83 of the manuscript, is a transcriptomic foundation model using pseudo-bulk pairs of chromatin accessibility and sequence data, differing in resolution and modality from ChromFound.
>
> In response to your professional suggestion, we will add citations to Epifoundation and ATAC-Diff, and update Section 2 to include a more thorough comparison. In addition, we clarify the positioning of ChromFound as the first genome-wide, self-supervised foundation model for scATAC-seq with zero-shot generalization across tissues.
>
> ## [Q3] Performance Variation Across Tissues
>
> Thank you for the insightful question. We have thoroughly evaluated ChromFound across various tissue types on all downstream tasks. While minor variations in performance exist due to inherent differences in tissue complexity and data sparsity, ChromFound consistently achieves strong results across all tasks and tissue types. If any specific aspect of the results remains unclear, please do not hesitate to let us know. We would be happy to provide further clarification.
>
> ## [Q4] Comparison Between WPSA and Linear Attention
>
> Thank you for the question. ChromFound’s WPSA is implemented using FlashAttention, significantly improving efficiency compared to vanilla self-attention. To further compare with linear attention, we implement Performer and Linformer under the same conditions, using identical hidden dimensions, batch size of 4, and FP16 mixed-precision training.
> The following results are obtained on the cell clustering task using the PBMC VIB_10xv1_1 dataset. As shown in the table below, WPSA outperforms both alternatives in terms of model performance and computational efficiency.
>
> |Model|Parameters|FLOPS| Inference speed(s/batch) |ARI(Trained on 0.02 million cells)|
> |-|-|-|-|-|
> |Ours|450305|7.88E+11|3.46|0.6142|
> |Performer|1838337|6.01E+12|3.89|0.5978|
> |Linformer|920833|1.62E+12|3.74|0.5751|
>
> ## [Q5] Statistical Significance and Random Seed Control
>
> We thank the reviewer for this professional and important question.
>
> For cell clustering (Table 1) and low-count denoising (Figure 2), we perform each evaluation 20 times with different random seeds and report the mean and standard deviation to reflect performance stability.
> In the perturbation response prediction task (Figure 4), we report p-values for the PCC metric to establish the statistical significance of the predictions.
> For other tasks, we ensure that all benchmark methods are implemented and evaluated under the same random seed settings for fair comparison.
>
> Together, these strategies ensure that all reported performance gains are both robust and statistically meaningful. We hope that all these clarifications have addressed reviewer's questions.
>
> ## [W6,L4] Ablation Study of less OCR
>
> We thank the reviewer for raising this important point. We try to understand the comment on "less OCR" in two possible ways, and we have conducted targeted ablation experiments to address both:
> 1. Reduce OCR input per cell: In Section 4.4 (Question 3), we reduce the maximum number of OCRs per cell to one-half and one-quarter of the full set by highly variable peaks filtering. We observe a clear performance drop in both clustering and annotation tasks, demonstrating that genome-wide OCR input is critical for capturing rich regulatory information and ensuring robust generalization.
> 2. Reduce pretraining data size: In Section 4.4 (Question 2), we also perform controlled experiments by reducing the pretraining corpus to one-tenth and one-hundredth of the original size. We observe a progressive decline in downstream performance, confirming the benefit of scaling up training data size.
>
> ## [L3] More Downstream Tasks
>
> We sincerely thank the reviewer for this suggestion. Actually, we have performed perturbation response prediction and presented the results in Figure 4. As the reviewer mentioned, temporal scATAC-seq prediction is quite challenging, and we are currently working on extending ChromFound to support this task.
>
> ## Conclusion
>
> We sincerely thank the reviewer for your valuable time and constructive feedback. We hope our responses have addressed your concerns. If there are any further questions, we would be happy to provide further clarification.

---

> ### Author Response · Authors · 2025-08-04
> **We Appreciate the Comments and Welcome Further Discussion**
>
> Dear reviewer:
>
> We sincerely thank the reviewer for your detailed and thoughtful feedback. Your comments have helped us better articulate the design motivations behind ChromFound, clarify technical choices, and position our work within the broader context of scATAC-seq modeling. We appreciate your recognition of the strengths of our approach, as well as your critical perspective on its limitations.
>
> In this rebuttal, we have carefully addressed each of your concerns through additional experiments, expanded comparisons, and clearer explanations. We have also updated our discussion of related work to reflect your suggestions, and we will revise the manuscript accordingly. We hope these responses provide a clearer understanding of our contributions and the rationale behind our methodological decisions.
>
> If there are any remaining questions or further points for clarification, please do not hesitate to reach out. Thank you again for your time, expertise, and engagement.

---

> > ### Comment · Reviewer_VAFg · 2025-08-06
> >
> > Thank you for the authors’ response, which addresses several of my earlier concerns. However, I still have reservations regarding the unified token representation across different datasets. While it is possible to encode chromatin accessibility using chromosome IDs, start/end positions (on GRCh38), and continuous accessibility values, the varying fragment lengths across datasets may inherently reflect differences in signal strength or resolution. Additionally, as you mentioned, the overlap of OCRs across tissues is relatively low. This raises concerns that the model may struggle to capture or infer the appropriate granularity of regulatory elements, especially when token representations are not fully harmonized.
> >
> > Moreover, although the model is capable of predicting both non-zero and zero OCRs, the class imbalance remains a significant issue—most regions are labeled as zero. This could bias the model toward predicting zeros to minimize loss, potentially resulting in a trivial solution. Compounding this, many zero-labeled regions may not correspond to truly closed chromatin, but rather to low signal or dropout events (e.g., promoters active in rare cell types). Treating these ambiguous regions as negatives could penalize the model unfairly and degrade generalization performance.
> >
> > I am also confused about the reported trend in Figure 2, where performance appears to improve as more aggressive downsampling is applied. This seems counterintuitive and warrants further clarification—does this suggest that the model benefits from reduced data complexity, or are there other confounding factors at play?

---

> > > ### Author Response · Authors · 2025-08-06
> > > **Reply to Official Comment by Reviewer VAFg (1/2)**
> > >
> > > # Concern 1: Unified Token Representation
> > > We thank the reviewer for highlighting this important concern. Traditional scATAC-seq studies often rely on a fixed vocabulary of OCRs (e.g., from a reference atlas) to harmonize scATAC-seq data across datasets. While this offers alignment consistency, it introduces two major limitations:
> > > 1. a loss of OCR length information, since OCR length inherently reflects differences in signal strength or resolution,
> > > 2. limited generalizability, as novel or tissue-specific OCRs may be excluded.
> > >
> > > These limitations restrict downstream applications, particularly in tissues with low overlap or previously unseen regulatory regions. To address this, ChromFound introduces a genomic-coordinate-based tokenization strategy that dynamically represents OCRs using their actual start and end positions. This approach explicitly encodes both the genomic span and location of each OCR, enabling the model to adapt to variable-length OCRs and capture regulatory heterogeneity across tissues.
> > >
> > > Theoretically, our unified token representation is designed to resolve two core challenges:
> > > 1. Varying fragment lengths: By incorporating start and end coordinates via sinusoidal positional embeddings (Section 3.1), the model learns from the complex extent of each OCR.
> > > 2. Low OCR overlap across tissues: Without relying on a fixed OCR vocabulary, ChromFound can flexibly accommodate tissue-specific or novel OCRs, mitigating misalignment and dropout issues.
> > >
> > > ChromFound further harmonizes scATAC-seq data by:
> > > - Applying log-transformation and dataset-specific normalization to accessibility values, which reduces inter-dataset signal discrepancies.
> > > - By retaining genomic coordinates during masked OCR reconstruction, ChromFound learns the relative spatial relationships between OCRs, which facilitates its adaptation to varying fragment lengths.
> > > - Training on a large and heterogeneous corpus (>1.9M cells across 30+ tissues), promoting generalization across a wide spectrum of fragment lengths and OCR compositions.
> > >
> > > Empirically, this design yields superior performance across various tissue types. We evaluate ChromFound on cPeaks, a standardized reference from "_A generic reference defined by consensus OCRs for scATAC-seq data analysis_", which defines 1,657,194 peaks across the human genome. We use a 20,000-cell PBMC subset for training and 8 evaluation datasets from various tissues in Table 1. As shown in the table below, ChromFound (Ours) consistently outperforms the version trained on the aligned datasets with cPeaks (Ours with cPeaks). These results demonstrate ChromFound's strong generalization ability across tissue types (e.g., 30% of peaks in retina, 35% in cortex, and 70% in heart overlap with the cPeaks-aligned PBMC training peaks) and its capability to handle varying OCR lengths (mean OCR lengths range from 500 to 1321 across datasets).
> > >
> > >
> > > |Dataset|Model (Trained on 0.02 million cells)|ARI|FMI|NMI|AMI|
> > > |-|-|-|-|-|-|
> > > |Cortex(batch 1)|Ours with cPeaks|0.5858|0.7244|0.7519|0.7499|
> > > ||Ours|0.6475|0.7503|0.7701|0.7689|
> > > |Cortex(batch 2)|Ours with cPeaks|0.5751|0.6720|0.7145|0.7123|
> > > ||Ours|0.6092|0.7015|0.7276|0.7259|
> > > |Heart(av 3)|Ours with cPeaks|0.5400|0.6347|0.6588|0.6567|
> > > ||Ours|0.5642|0.6601|0.6981|0.6953|
> > > |Heart(av 10)|Ours with cPeaks|0.6150|0.7695|0.6968|0.6963|
> > > ||Ours|0.6432|0.7869|0.7220|0.7223|
> > > |Retina(D026_13)|Ours with cPeaks|0.4888|0.7081|0.7241|0.7238|
> > > ||Ours|0.5802|0.7549|0.7410|0.7403|
> > > |Retina(D19D008)|Ours with cPeaks|0.5368|0.6839|0.7664|0.7660|
> > > ||Ours|0.5956|0.7202|0.7832|0.7868|
> > > |PBMC(VIB_10xv1_1)|Ours with cPeaks|0.5999|0.6827|0.7434|0.7424|
> > > ||Ours|0.6142|0.6995|0.7354|0.7345|
> > > |PBMC(BIO_ddseq_1)|Ours with cPeaks|0.4347|0.6237|0.5867|0.5861|
> > > ||Ours|0.4579|0.6421|0.5902|0.5910|
> > >
> > > In summary, building a foundation model with position-aware, unified OCR representation enables robust modeling across tissues and unlocks broader applicability compared to approaches based on pre-defined OCR references. We hope our detailed explanation and empirical evidence help address your concerns.

---

> > > ### Author Response · Authors · 2025-08-06
> > > **Reply to Official Comment by Reviewer VAFg (2/2)**
> > >
> > > # Concern 2: Dropout Effect
> > > ## Concern 2.1: Zero and Non-zero OCRs Imbalance
> > > We appreciate the reviewer’s insightful concern. To mitigate the imbalance between zero and non-zero OCRs, we design our pretraining strategy with the following considerations:
> > > 1. Balanced masking: We apply a symmetric masking strategy that samples equal numbers of zero and non-zero OCRs during pretraining. This ensures that the model learns from both accessible and inaccessible regions, thereby avoiding class imbalance in the reconstruction objective.
> > > 2. Normalized regression loss: Instead of predicting raw count data, we apply a log-transform followed by dataset-specific normalization to the accessibility values. We then use MSE as the reconstruction loss. This loss function improves stability and avoids the sensitivity of Poisson or ZINB losses to extreme count variability across datasets.
> > >
> > > We hope this clarifies how our design addresses the potential bias introduced by the imbalance of zero and non-zero OCRs. Please let us know if any concerns remain.
> > > ## Concern 2.2: Low Signal or Dropout Events instead of Truly Closed Chromatin
> > > We fully agree that many zero-labeled OCRs may result from dropout or insufficient coverage, especially for active regions in rare cell types. This ambiguity is a known limitation of scATAC-seq and is inherently difficult to resolve within any single dataset.
> > >
> > > ChromFound does not rely on individual sample-level ground truth to resolve ambiguous zeros. Instead, it captures population-level statistical patterns from diverse datasets, enabling it to implicitly distinguish consistent regulatory signals from dataset-specific sparsity or noise. Our comprehensive experiments (Table 1, Fig.2, Table 2, Fig.3, Table 3) demonstrate that ChromFound achieves strong performance across diverse tissues and tasks.
> > > For example, a promoter that is dropped out in one dataset may still be reliably learned from others where it is consistently detected. This cross-context learning is a key strength of foundation models, capturing robust biological signals beyond dataset-specific artifacts.
> > >
> > > We hope this explanation addresses your concern regarding dropout-induced zeros and demonstrates how ChromFound mitigates such ambiguity through large-scale cross-context learning. We welcome any further questions or suggestions.
> > >
> > > # Concern 3: Improvements as more aggressive downsampling
> > > We thank the reviewer for raising this point. To clarify, this trend is only observed in the retina datasets in Figure 2, not across all tissues. Our analysis, shown in the following table, suggests that this is due to differences in sequencing protocols. Retina datasets contain a higher proportion of OCRs that are frequently accessible across cells. These high-frequency OCRs likely reflect technical redundancy or sequencing noise, which reduces the signal-to-noise ratio.
> > >
> > > |Dataset|#Cells|#Peaks|% OCRs open in ≥20% of cells|% OCRs open in ≥10% of cells|% OCRs open in ≥5% of cells|
> > > |-|-|-|-|-|-|
> > > |Retina(D19D008)|4575|232354|**0.36**|**4.13**|**10.76**|
> > > |Cortex(batch 2)|1416|212793|0.02|0.38|2.61|
> > > |Heart(av 3)|2785|429828|0.15|1.60|4.42|
> > > |PBMC(BIO_ddseq_1)|5708|412490|0.21|2.27|5.17|
> > >
> > > In this case, random downsampling (via Bernoulli sampling) suppresses these abundant OCRs, effectively reducing noise and enhancing signal clarity for downstream representation learning. This explains the observed performance improvement in the retina dataset in Figure 2.
> > >
> > > Thank you for the professional suggestion. We will clarify this point in the revised manuscript. If you have any further questions, please do not hesitate to let us know.

---

> > > > ### Comment · Area_Chair_X8gW · 2025-08-08
> > > >
> > > > Hi reviewer VAFg,
> > > >
> > > > Thanks for your engagement in the lively discussion. If you would like to respond to the authors' most recent messages, please do so by end of the day today AOE.

---

### Official Review · Reviewer_2Z9j · 2025-07-03

**Clarity:** 4
**Significance:** 4
**Originality:** 3
**Rating:** 6
**Confidence:** 4

**Summary:**

The authors proposed ChromFound, an encoder-decoder framework to build foundation model for scATAC-seq data. Genomic coordinate information and chromatin accessibility profile were used to represent open chromatin regions (OCR). The authors proposed window partition self-attention (WPSA) module to capture local dependencies among OCRs, and Mamba layer to handle ultra-long OCRs. The decoder is MLP layer.
ChromFound shows improved performance on cell representation accommodating low count and batch effect, which are popular in scATAC-seq data. ChromFound also improves cell type annotation and cross -omics data prediction.Application of ChromFound on enhancer-gene link prediction and perturbation response prediction shows the results make biological sense.

**Questions:**

1. it is interesting to see how will it perform on gene regulatory network prediction?
2. A nature way is to incorporate DNA sequence  and or scRNA-seq information into the model, since DNA sequence is already there and scRNA-seq data is more widely available.
3. minor comments:
   reference to Table 4.4 should be Table 4
   In figure 6, the order of figure in Harmony panel is reversed compared with other methods.

**Ethical Concerns:**

["NO or VERY MINOR ethics concerns only"]

**Final Justification:**

The authors have provided insights on future work on incorporating DNA sequence, scRNA-seq, and gene regulatory network to address my comments.

**Limitations:**

1. The authors forget to include limitations, such as the model is only on human data, which can be extend to include mouse and other species

**Quality:**

4

**Strengths And Weaknesses:**

The submission is technically sound.  The experimental results are well supporting the authors' claim. The methods used are appropriate. The work is almost complete piece of work. The authors mentioned strengths of their work carefully and honestly.

The submission is clearly written and well organized. It adequately inform the reader. No source code was provided.

The authors propose for the first time foundation model for scATAC-seq and achieve promising results compared with existing approaches.  It is very likely other researchers will utilize or extend the approach to address more challenges in scATAC-seq data analysis.

The authors offer a novel combination of existing techniques, and the reasoning behind this combination is well-articulated. The work addresses challenges in generating foundation model in scATAC-seq with suitable techniques and adaption of existing technique when needed. The authors clearly distinguish their contributions with previous contributions.

---

> ### Author Rebuttal · Authors · 2025-07-31
>
> We sincerely thank you for your thoughtful and constructive feedback. We are encouraged by your recognition of the technical soundness, comprehensive evaluation, and potential impact of ChromFound. Below we address your comments in detail.
>
> ## [W1] Source Code
> We appreciate your comment regarding code availability. We are fully committed to promoting transparency and reproducibility. Upon acceptance, we will release the complete source code, pretrained weights and tutorials for all experiments.
>
> ## [Q2] DNA sequence modeling
> Thank you for this insightful suggestion. We totally agree that DNA sequence information can complement chromatin accessibility data by providing base-resolution regulatory context. We note that scBasset also incorporates local DNA sequence to predict binary chromatin accessibility and has demonstrated strong performance in cell representation and batch correction.
>
> While ChromFound currently does not explicitly encode DNA sequence, our tokenization schema can be extended to include local sequence features around each OCR. This would allow the model to jointly learn from static genomic sequences and dynamic chromatin accessibility profiles.
> One possible way of integrating DNA sequence is to fuse the DNA sequence embeddings from DNA foundation models (e.g., Evo2, Nucleotide Transformer, Enformer, etc.) into the ChromFound architecture. This approach would allow ChromFound to benefit from both dynamic chromatin activity and static regulatory code, potentially improving its ability to resolve fine-grained enhancer-gene interactions. We will explore this direction in future work.
>
> ## [Q2] scRNA-seq modeling
> Thank you for your professional suggestion. We totally agree that incorporating scRNA‑seq is both natural and valuable. Many current methods jointly model scATAC‑seq and scRNA‑seq by using scATAC‑seq as input to predict scRNA‑seq expression (e.g., GET) or perform multi‑omics integration through shared embeddings (e.g., GLUE, scMODAL).
>
> However, these approaches typically treat peaks and genes as separate modalities linked only in downstream alignment. A current limitation in existing approaches is the lack of modeling under a unified genomic scale. We hope that the unified genomic scale will place peaks and genes on the same coordinate framework to enable end‑to‑end training from the genome sequence to transcriptomic output.
>
> In our future work, we aim to bridge this gap by pretraining ChromFound in a truly genomic-centered manner: jointly embedding OCRs and genes in the same latent space from the start, enabling the model to internalize the central dogma (i.e., the information flow from DNA to RNA to protein) via self‑supervised objectives. While paired single-cell scATAC‑seq and scRNA‑seq data at sufficient scale remain limited for pretraining, we are actively pursuing large‑scale collection and simulation strategies to realize this vision.
>
> ## [Q1] GRN prediction
> We appreciate the reviewer’s suggestion on GRN prediction. ChromFound is currently designed to infer _cis_-regulatory interactions, such as enhancer-gene links and perturbation responses. Looking forward, we aim to expand ChromFound to support broader gene regulatory network (GRN) inference.
>
> As discussed in the response regarding scRNA-seq modeling, we envision extending ChromFound to jointly model scATAC-seq and scRNA-seq data in a truly genomic-centered manner. This unified framework would allow the model to capture not only local _cis_-regulatory dependencies, but also _trans_-regulatory interactions.
> While data limitations currently constrain pretraining across paired multi-omics at single-cell resolution, we are actively exploring strategies to realize this integrative modeling paradigm in future work.
>
> ## [Q3] Minor comments
> Thank you for pointing this out. We will correct the table reference to Table 4 and revise Figure 6 in the revised manuscript to maintain consistent panel ordering.
>
>
> ## [L1] Limitations on Human Data
> We appreciate your suggestion to more explicitly discuss limitations. While we acknowledge that mouse datasets are often more abundant and widely available, our motivation is to prioritize human data due to its direct relevance for studying disease mechanisms and interpreting non-coding variants associated with human traits. We believe that this choice has greater significance for advancing biomedical research and understanding human-specific regulatory programs.
>
> Nevertheless, extending ChromFound to other species is an important future direction. Our genome-aware tokenization is inherently adaptable, but cross-species modeling will require additional algorithmic revisions, such as species-aware OCR tokenization and the alignment of genomic coordinates across organisms. We plan to explore multi-species pretraining and cross-species transfer learning in future work.
>
> ## Conclusion
> Once again, we truly thank the reviewer for the valuable comments and strong endorsement of our work. We hope our responses have addressed your concerns, and we are happy to provide further clarifications if needed.

---

> > ### Comment · Reviewer_2Z9j · 2025-08-08
> >
> > Thanks authors for your response and addressing the comments.

---

### Official Review · Reviewer_ybr7 · 2025-07-05

**Clarity:** 3
**Significance:** 3
**Originality:** 4
**Rating:** 5
**Confidence:** 4

**Summary:**

This paper presents ChromFound, the first foundation model developed specifically for single-cell chromatin accessibility (scATAC-seq) data. The authors propose a novel hybrid architecture that leverages a Mamba block for long-range genomic context and a windowed self-attention (WPSA) module for local regulatory interactions. A core contribution is the genome-aware tokenization scheme that encodes genomic coordinates and continuous accessibility values, addressing the challenge of heterogeneous data inputs. After pre-training on a large-scale corpus of 1.97 million human cells, the model is comprehensively evaluated across six downstream tasks, where it demonstrates state-of-the-art performance and notable zero-shot capabilities.

**Questions:**

1.  Could you elaborate on the design choice to project to a lower-dimensional space ($D_{low}=32$) for the Mamba block? What were the primary trade-offs you considered between performance and computational efficiency (e.g., parameter count, inference speed)?
2.  The zero-shot performance for cell representation is quite strong, yet fine-tuning yields considerable gains for tasks like cell type annotation. What is your interpretation of this? Does it suggest that the pre-trained model captures general cell state well, but that cell-type-specific regulatory grammars are only fully resolved during task-specific fine-tuning?
3.  Regarding the positional embeddings, could you comment on the model's sensitivity to the `temp` hyperparameter? Is performance stable across a range of values, or was this a carefully tuned parameter?
4.  The fixed window for WPSA is well-justified for typical enhancer interactions. Have you explored its potential limitations for capturing regulatory phenomena over larger genomic distances, and is the Mamba block intended as the primary mechanism for these cases?
5.  Given the pre-training corpus, what are your expectations for ChromFound's zero-shot or few-shot performance on cell types from disease contexts not well-represented in the training data, such as autoimmune disorders?

**Ethical Concerns:**

["NO or VERY MINOR ethics concerns only"]

**Final Justification:**

Authors have effectively addressed the key issues I raised:

**Hyperparameter Analysis**: The systematic evaluation of both the positional embedding temperature and Mamba projection dimension provides exactly the type of sensitivity analysis I was looking for. The clear trade-off between performance and computational efficiency (particularly the memory constraints at d_proj=128) gives readers valuable guidance for practical implementation.

**Enhanced Ablation Study**: Your comparison with pure Mamba and the creative approach to Transformer baselines (using Geneformer and scGPT as proxies) convincingly demonstrates that the performance gains stem from your hybrid architectural design, not merely from large-scale pretraining. The ~15% ARI drop when removing WPSA and the fundamental limitations of existing Transformer-based models on this data modality strengthen your architectural claims significantly.

**Robustness and Generalization**: Your clarification on coordinate-based tokenization and the existing disease representation in your evaluation datasets addresses my concerns about generalizability. The downsampling experiments provide good evidence for robustness to preprocessing variations.

The addition of computational profiling results also addresses practical deployment considerations, which will be valuable for the research community.

Your rebuttal demonstrates not only technical rigor but also a deep understanding of the biological context and practical implications of your work. The responses show that ChromFound represents a well-engineered solution to genuine challenges in single-cell chromatin accessibility analysis.

**Updated Rating: 5 (Accept)** - This is technically sound work with clear contributions that will be valuable to the research community. The comprehensive evaluation and your responsive improvements to the analysis strengthen confidence in the work's impact and reliability.

**Limitations:**

yes

The authors appropriately acknowledge the model's human-centric focus as a primary limitation. For a more comprehensive discussion, I would suggest also including a brief commentary on two other practical points: (1) the model's implicit reliance on the quality of user-provided peak calls, which is an important contextual factor for anyone applying the model, and (2) the practical compute resources required for fine-tuning, which is relevant for assessing the model's accessibility to the broader research community.

**Paper Formatting Concerns:**

No concerns

**Quality:**

4

**Strengths And Weaknesses:**

**Strengths**

* The work successfully establishes the first foundation model for a challenging and important biological data modality. This is a novel and impactful contribution that charts a new direction for the field.
* The scale of the pre-training is impressive and a key factor in the model's strong performance. The subsequent evaluation is both rigorous and comprehensive, providing convincing evidence for the model's capabilities across a wide array of relevant tasks.
* The hybrid Mamba-WPSA architecture is not merely a concatenation of popular methods; it's a well-justified design choice that reflects the multi-scale nature of genomic regulation, which is a commendable feature.

**Weaknesses**

While the work is strong overall, there are several areas where a deeper analysis would strengthen the paper's conclusions and provide a more complete picture for the research community. I've ordered these from minor points to more substantial considerations.

* On a minor note, while key hyperparameters are provided, the paper would benefit from a brief discussion on the model's sensitivity to these choices (e.g., the `temp` value in positional embeddings or the Mamba projection dimension). This would provide readers with a better sense of the model's robustness and tuning requirements.
* A more significant point is that the ablation study, while useful, could be more comprehensive. The current study primarily demonstrates that removing major architectural blocks is detrimental. However, it leaves open important questions about the sources of performance gain. For instance, a comparison to simpler, strong baselines (like a pure Mamba or a pure Transformer architecture) trained on the same massive dataset would be invaluable. Such an analysis would help disentangle the benefits of the proposed hybrid design from the undeniable power of the large-scale pre-training data.
* Finally, the most substantial area for improvement relates to the model's generalizability and its dependency on upstream analytical choices. The model's performance is intrinsically linked to the input set of Open Chromatin Regions (OCRs), which can vary significantly based on the peak-calling algorithm used. A sensitivity analysis or at least a discussion on this crucial variable would be necessary to fully substantiate the model's claim of universality. Similarly, the model's impressive performance may not fully generalize to disease contexts (e.g., autoimmune disorders) that are not well-represented in the pre-training corpus, a key consideration for its application as a foundational tool for discovery.

---

> ### Author Rebuttal · Authors · 2025-07-31
>
> We sincerely thank the reviewer for the thoughtful and detailed feedback. We appreciate your recognition of ChromFound’s novelty, the biologically informed architectural design, and the breadth of downstream evaluations. Below, we address each of your suggestions and concerns.
>
> ## [W3,L1] Discussion on Peak-calling Methods
> Thank you for the important point. We agree that different upstream data pipelines can lead to variability in the cell-by-peak matrix and peak sets, particularly in **sparsity** and **peak boundary definitions**. ChromFound is designed to be robust to these variations.
>
> - To assess robustness to such sparsity differences, we perform downsampling experiments (Figure 2) simulating varying levels of accessible peaks per cell. ChromFound maintains stable performance across all conditions, indicating strong resilience to sparsity variation introduced by upstream preprocessing.
>
> - To address potential peak shift caused by different peak calling methods, ChromFound encodes OCRs using sinusoidal positional embeddings of their genomic coordinates. This design enables the model to capture relative spatial relationships between peaks, enhancing robustness to peak boundary shifts. Unlike dictionary-based approaches, our coordinate-based tokenization reduces reliance on exact peak definitions. We indeed conduct experiments aligning peaks across datasets to validate ChromFound’s robustness to peak shift. Due to space limitations, we kindly refer the reviewer to our response to Reviewer dMbP’s comment 2 for detailed results. We appreciate your understanding.
>
> In summary, while the pretraining and evaluation datasets are from different upstream analytical tools, ChromFound still achieves strong performance and generalization. We will include a discussion of these points in the revised manuscript. Please feel free to discuss further if you have additional concerns.
>
> ## [W3,Q5] Generalization to Disease Contexts
> We appreciate your interest in ChromFound's generalization to disease contexts. As detailed in Appendix B Table 5, our pretraining corpus does span **6 disease types**, including Alzheimer’s, Parkinson’s, leukemia, glioma, myocardial infarction and breast cancer.
>
> Moreover, we have already included two disease benchmark datasets in downstream tasks: Morabito130K, derived from **Alzheimer’s disease**, and Kuppe139K, a heart dataset from patients with **myocardial infarction**. ChromFound demonstrates strong performance on both datasets (Table 1, Figure 2, Table 2), supporting the model’s robustness in disease contexts.
>
> We hope these results address the reviewer’s concerns. In addition, We have not found a scATAC-seq dataset of autoimmune disorders with explicit cell labels. If the reviewer has specific datasets in mind, we would be happy to explore them further.
>
> ## [W2] More Comprehensive Ablation Study
> Thank you for the insightful suggestion. As requested, we compare ChromFound to both a pure Mamba and a pure Transformer baseline under the same pretraining corpus.
> 1. The pure Mamba model (Row 3 of Table 4) removes WPSA and uses Mamba alone. This results in a ~15% ARI drop, confirming that local attention is critical and that the hybrid design contributes synergistically to performance.
> 2. Training a vanilla Transformer on scATAC-seq inputs containing nearly one million peaks is practically infeasible due to its quadratic scaling in memory and computation. To approximate a pure Transformer baseline under practical settings, we train two representative single-cell foundation models, Geneformer and scGPT, on the same pretraining corpus with increasing peak lengths (4k–32k) and the same hyperparameters as WPSA. The results of comparison on the cell clustering task using the PBMC169K (batch VIB_10xv1_1) dataset are detailed in the table below.
>     - Geneformer sorts peaks by accessibility value before input, which disrupts the native genomic order. As a result, it fails to learn meaningful representations.
>     - scGPT applies a highly variable OCR selection strategy, limiting inputs to a small subset of peaks. Performance saturates at ARI ≈ 0.39, constrained by the maximum input length that Transformers can handle efficiently.
>
> |Method|Peak Length|ARI|
> |-|-|-|
> |Geneformer|4096|0.0451|
> ||16384|0.0457|
> ||32768|0.0460|
> |scGPT|4096|0.3075|
> ||16384|0.3774|
> ||32768|0.3868|
> |ChromFound|363066|0.6953|
>
> ## [W1,Q1,Q3] Model Sensitivity to Hyperparameters
>
> Thank you for this helpful suggestion. We have evaluated both hyperparameters during early-stage model development on the cell clustering task using PBMC169K (batch VIB_10xv1_1) dataset. The results and findings are summarized below.
>
> 1. Positional embedding temperature: This parameter controls the frequency of sinusoidal position encodings and can be viewed as a proxy for “genomic resolution.” As shown below, performance slightly drops when temp is within 1e3 to 1e5, and degrades more substantially when temp is too large, likely due to loss of relative position sensitivity.
>
> |$temp$|ARI|NMI|AMI|FMI|
> |-|-|-|-|-|
> |1000|0.6535|0.7764|0.7755|0.7259|
> |10000|0.6512|0.7709|0.7701|0.7243|
> |100000(Ours)|0.6953|0.7860|0.7852|0.7601|
> |1000000|0.6036|0.7344|0.7334|0.6860|
> |10000000|0.5852|0.7269|0.7258|0.6714|
>
> 2. Mamba projection dimension: This parameter controls the compression within the Mamba block. As shown below, $D_{low}$ = 32 achieves a strong balance between performance and efficiency. Our choice of 32 is primarily motivated by the trade-off between performance and computational efficiency. Larger values lead to marginal gains but significantly increase FLOPs and memory, with $D_{low}$ = 128 exceeding the memory limits on A100 80G GPUs.
>
> |$D_{low}$|Parameter Count|FLOPs|Inference Speed (s/sample)| GPU memory (GB) |ARI|
> |-|-|-|-|-|-|
> |16|422,593|7.41E+11|3.4027|48.8|0.6158|
> |32(Ours)|450,305|7.89E+11|3.4624|60.7|0.6953|
> |64|518,785|9.09E+11|4.0053|72.3|0.6927|
> |128|707,969|1.24E+12|OOM|OOM|OOM|
>
> We hope these results address the reviewer's concerns. We will include this analysis in the revised manuscript.
>
> ## [Q2] Interpretation of Fine-tuning Benefits
>
> Thank you for the thoughtful and professional insight, and we strongly agree with your interpretation. The strong zero-shot performance indeed suggests that ChromFound learns generalizable representations of cellular states and underlying regulatory structures during pretraining. These representations are already informative for unsupervised clustering and can reveal biologically relevant subpopulations.
>
> As you rightly pointed out, fine-tuning further improves performance by enabling the model to specialize in cell-type-specific regulatory features. Such features often involve subtle patterns or rare combinations of regulatory elements that are better resolved with supervision. This fine-tuning complements the global regulatory landscape already captured during pretraining. In addition, we have provided a detailed analysis of the cell type annotation confusion matrix in Appendix F.2, demonstrating ChromFound’s advantage in recognizing rare and long-tail cell types and cell-type-specific regulatory patterns.
>
> We appreciate your expert perspective and will incorporate this interpretation into our revised discussion. Your feedback greatly enhances the clarity of our presentation and strengthens our understanding of how pretraining and fine-tuning synergize in ChromFound.
>
> ## [Q4] Fixed WPSA Window
> Thank you for raising this important point. Both Hi-C[1] and CRISPRi[2] have shown that the vast majority (over 90%) of validated enhancer-gene interactions occur within 200 kb of the transcription start site (TSS). Theoretically, the Mamba block modeling genome-wide dependencies allows ChromFound to infer potential distal enhancer-gene links. That said, we acknowledge that experimental validation of distal enhancer–gene links (>200 kb) remains limited, we view this as an important direction for future work. If the reviewer is aware of suitable datasets, we would be happy to incorporate such experiments.
>
> [1] Bonev, B. & Cavalli, G. Organization and function of the 3D genome. Nat Rev Genet 17, 661–678 (2016).
>
> [2] Fulco, C. P. et al. Activity-by-contact model of enhancer–promoter regulation from thousands of CRISPR perturbations. Nat Genet 51, 1664–1669 (2019).
>
> ## [L2] Compute Resources for Fine-tuning
> We thank the reviewer for highlighting the importance of computational efficiency. We provide profiling results on cell type annotation and cross-omics prediction tasks using a single A100 80G GPU:
>
> |Task|Dataset|Peak Length|Batch Size|Training Speed (s/step)|Peak Memory Usage (GB)|
> |-|-|-|-|-|-|
> |Cell Type Annotation|Bone 43/44|495,416|4|0.535|67.8|
> |Cross-omics Prediction|BMMC multiome 2021|269,544|4|1.060|56.7|
>
> We note that the primary driver of memory consumption is the length of the input peak sequence, which can reach several hundred thousand tokens per cell. In contrast, the model architecture itself remains lightweight (parameter count: 450305). All current resource metrics will be included in the appendix to guide users in estimating compute requirements. In future work, we aim to further optimize memory efficiency and training speed for broader applications.
>
> ## Conclusion
> We hope these responses address your concerns and questions. We sincerely appreciate your valuable feedback and constructive suggestions. If you have any additional questions or suggestions, please feel free to let us know.

---

> ### Author Response · Authors · 2025-08-04
> **We Appreciate the Suggestions and Welcome Further Discussion**
>
> Dear reviewer:
>
> We sincerely thank the reviewer for the thoughtful, detailed, and constructive feedback. We appreciate your recognition of ChromFound’s novelty, biologically grounded architecture, and the comprehensive downstream evaluations. In this rebuttal, we have carefully addressed each of your suggestions, including analysis of upstream dependencies, generalization to disease contexts, expanded ablation studies, and sensitivity to key hyperparameters.
>
> We hope our responses have addressed your concerns and clarified the rationale behind our design choices. If any questions remain or if you would like to discuss any points further, we would be very glad to continue the conversation. Your feedback has been instrumental in strengthening our work, and we are truly grateful for your time and insight.

---

### Official Review · Reviewer_dMbP · 2025-07-07

**Clarity:** 3
**Significance:** 2
**Originality:** 3
**Rating:** 4
**Confidence:** 4

**Summary:**

ChromFound is a foundation model for single-cell ATAC-seq data pretrained on ~2 million cells that aims to learn batch corrected representations of cells that (a) cluster by cell type, (b) can be used to predict cell types for new cells, and (c) can be used to predict other modalities (such as RNA-seq). Across all three aforementioned tasks, the authors show that ChromFound outperforms baseline methods that train on specific datasets and are not pre-trained.

The model is pre-trained in a self-supervised fashion to predict masked accessibility values. Crucially, the model parametrizes open chromatin regions (OCRs) by their start and end location, as opposed to having a trainable embedding for a specific OCR, allowing the model to work on OCRs from new datasets, which will not exactly match OCRs from the pre-training datasets.

**Questions:**

Major suggestions/questions:
1. Please compare to non-pretrained ChromFound for all experiments, not just Fig. 3, Table 3, and Table 4.
2. Benchmark against Signac (TF-IDF + LSI), peakVI/poissonVI, and SnapATAC2 for cell clustering tasks, as these are the methods most commonly used by practioners. For each, the appendix should indicate the exact pre-processing done before applying the method (e.g. are peaks filtered by their observation frequency before passing them into peakVI?).
3. Do not use scVI/scANVI for batch-effect correction, as they are designed for RNA-seq. Use peakVI/poissonVI instead.
4. What input representations do you pass to Harmony for batch-effect correction? PCA or LSI or neither?
5. What l2 regularization parameter do you use for scBasset for batch-effect correction? How do you choose this regularization parameter?
6. A small linear layer on top of low-dimensional representations learned by Signac, peakVI, and SnapATAC2 can also be used for cell type annotation. Please compare to those baselines.


Minor suggestions/questions:
1. At test time, when cell type labels are unavailable, how does OCR filtering (as described on lines 611-612) work?
2. Why do you use an MSE loss after log transformation instead of a Poisson loss on the raw counts, which likely better fits the generative process.
3. In Fig. 4, are you using ABC predictions as ground-truth? This seems suspect as ABC itself is far from perfect and has several biases (i.e. overly discounting distal enhancers).
4. Question 3 in the ablations asks if long-context modeling using Mamba layers helps, but I'm not sure if the experiment performed actually answers that question. My reading of Table 4 is that half and three-fourths of the OCRs are just removed from the input data, but that doesn't test the utility of the Mamba layers directly. A better test would be to use all OCRs, but pool after the WPSA layer without having any Mamba layers.

**Ethical Concerns:**

["NO or VERY MINOR ethics concerns only"]

**Final Justification:**

I've increased my score from a 3 to a 4 based on the authors re-running of baselines with more suitably chosen hyperparameters. It seems like this method is at least as good as existing methods. And even if further hyperparameter optimization of the baselines reveals that this method is only slightly better than them (which I suspect to be the case), it at least offers a novel and interesting approach to merge OCRs from different datasets that will be valuable to the single-cell ATAC-seq community.

**Limitations:**

Yes

**Quality:**

2

**Strengths And Weaknesses:**

To my knowledge, ChromFound is the first foundation model for scATAC-seq data. In my opinion, the paper's primary contribution is the demonstration that pre-training on millions of scATAC-seq cells enables zero-shot representation learning for completely new datasets, without requiring additional training or fine-tuning. (I do not think the model itself minus the pre-training is a general advance given the results in Fig. 3 and Table 3, which show that baselines are competitive with a non-pretrained ChromFound model.) This type of pre-training has mostly not been done before because different datasets have different peak/OCR boundaries and harmonizing peak sets is not straightforward. The authors cleverly solve this problem by parametrizing OCRs by a function of their start and end positions, as opposed to using an embedding table. This I find to be their primary computational contribution.

However, there are other methods of harmonizing peak sets that would enable other architectures (such as a standard VAE) to be used for pre-training that the authors do not compare to. Computational biologists routinely (i) aggregate fragment files from multiple datasets to re-call a unified set of peaks or (ii) adopt the peak set of a large reference atlas. If fragment files are available for the new dataset, then peak counts can be called specifically against the reference peak set. But even if fragment files are not available and only a count matrix is, counts from the new peak set can be transferred to the reference peak set by examining peak overlaps. Demonstrating that ChromFound outperforms (a) both itself with a different harmonization strategy and (b) other methods pre-trained using one of the aforementioned harmonization strategies would substantially strengthen the paper.

The other notable weakness of the paper is the benchmarking to baselines. The authors omit several standardly used tools, do not systematically compare against the same tools for all tasks even though most methods are generally applicable, and crucially do not detail how they implement the baselines. For cell clustering and batch-effect correction, Signac, peakVI, and SnapATAC2 are the most commonly used tools and should be compared against.  For cross-omics predictions, Signac and multiVI should also be compared against. Additional recommendations are listed in the "Questions" section.

---

> ### Author Rebuttal · Authors · 2025-07-31
>
> We appreciate the reviewer for recognition of ChromFound’s novelty and constructive feedbacks. Below, we summarize our key revisions and additional analyses in response:
>
> ## [1]. Benchmark completeness
>
> 1. We expand the clustering benchmarks (Table 1) by adding Signac (Sig), SnapATAC2 (SAT2), peakVI (pkVI), and poissonVI (psVI). All methods use the same preprocessing (Appendix C). Signac, peakVI, and poissonVI skip log normalization; scBasset uses binarized input.
>
> |Dataset|Model|ARI|FMI|NMI|AMI|
> |-|-|-|-|-|-|
> |Cortex(batch 1)|Sig|0.5013|0.7035|0.5580|0.5534|
> ||SAT2|0.5118|0.6740|0.7202|0.7179|
> ||pkVI|0.4558|0.6286|0.6964|0.6940|
> ||psVI|0.3673|0.5673|0.5878|0.5845|
> ||Ours|0.6890|0.7943|0.7779|0.7760|
> |Cortex(batch 2)|Sig|0.5069|0.6478|0.5888|0.5851|
> ||SAT2|0.5325|0.6380|0.7038|0.7015|
> ||pkVI|0.5512|0.6530|0.7050|0.7028|
> ||psVI|0.5334|0.6388|0.6004|0.5983|
> ||Ours|0.6278|0.7156|0.7321|0.7299|
> |Heart(av 3)|Sig|0.3123|0.5093|0.2294|0.2281|
> ||SAT2|0.4571|0.7285|0.6867|0.6863|
> ||pkVI|0.4159|0.6383|0.6149|0.6144|
> ||psVI|0.3969|0.5929|0.4472|0.4464|
> ||Ours|0.5828|0.6757|0.7207|0.7187|
> |Heart(av 10)|Sig|0.1159|0.4601|0.5272|0.5238|
> ||SAT2|0.5515|0.5564|0.6625|0.6502|
> ||pkVI|0.4301|0.5434|0.6024|0.6000|
> ||psVI|0.3606|0.5232|0.5717|0.5688|
> ||Ours|0.6774|0.8109|0.7369|0.7365|
> |Retina(D026_13)|Sig|0.5538|0.7451|0.4931|0.4919|
> ||SAT2|0.5606|0.7207|0.7045|0.7041|
> ||pkVI|0.2885|0.5543|0.5644|0.5638|
> ||psVI|0.3162|0.5766|0.6081|0.6076|
> ||Ours|0.6668|0.8149|0.7644|0.7641|
> |Retina(D19D008)|Sig|0.5146|0.7339|0.5235|0.5222|
> ||SAT2|0.5814|0.7173|0.7881|0.7877|
> ||pkVI|0.4702|0.6318|0.7229|0.7224|
> ||psVI|0.4978|0.6527|0.7400|0.7395|
> ||Ours|0.6688|0.7767|0.8183|0.8179|
> |PBMC(VIB_10xv1_1)|Sig|0.1832|0.4543|0.3064|0.3027|
> ||SAT2|0.6246|0.7085|0.7551|0.7541|
> ||pkVI|0.5417|0.6340|0.6639|0.6627|
> ||psVI|0.6314|0.7073|0.7353|0.7343|
> ||Ours|0.6953|0.7601|0.7860|0.7852|
> |PBMC(BIO_ddseq_1)|Sig|0.1885|0.5249|0.2852|0.2839|
> ||SAT2|0.4126|0.6149|0.5761|0.5754|
> ||pkVI|0.2915|0.5041|0.4623|0.4615|
> ||psVI|0.4103|0.5994|0.5656|0.5649|
> ||Ours|0.4835|0.6604|0.5950|0.5944|
>
> 2. We expand batch correction evaluation (Table 2), including peakVI (pkVI) and poissonVI (psVI) to replace scVI and scANVI.
>
> |Tissue|Model|AvgBIO|AvgBATCH|
> |-|-|-|-|
> |Bone|pkVI|0.2981|0.9173|
> ||psVI|0.2936|0.9178|
> ||Ours|0.6408|0.9289|
> |Heart|pkVI|0.5657|0.7564|
> ||psVI|0.5621|0.7719|
> ||Ours|0.8180|0.8679|
> |PBMC|pkVI|0.6172|0.7889|
> ||psVI|0.6029|0.8081|
> ||Ours|0.6443|0.8217|
> |Cortex|pkVI|0.7256|0.9254|
> ||psVI|0.7240|0.9279|
> ||Ours|0.7440|0.9565|
>
> 3. The descriptions for all benchmark methods are provided in Appendix H. The APIs and tools used are as follows:
>      - scBasset, peakVI, poissonVI, scVI: scvi.model
>      - Liger: the pyliger Python package
>      - Scanorama, Harmony: scanpy.external
>      - scANVI: scib.integration
>      - Cross-omics benchmarks: DANCE.modules.multi_modality.predict_modality
>      - Others: official source codes and tutorials from their respective publications
>
>     To directly address the reviewer’s question:
>      - L2 regularization parameter for scBasset is set to 1e-8;
>      - Input representation passed to Harmony is PCA.
>
> 4. We include logistic regression results in cell type annotation (Fig. 3) using Signac (Sig), peakVI (pkVI), and SnapATAC2(SAT2) embeddings. The results of baselines are much worse than ChromFound's.
>
> |Tissue|Train|Test|Method|Accuracy|F1|
> |-|-|-|-|-|-|
> |PBMC|EPF_hydrop_1|VIB_10xv1_1|Sig|0.3546|0.1643|
> ||||pkVI|0.2756|0.1430|
> ||||SAT2|0.1256|0.1751|
> |PBMC|EPF_hydrop_3|VIB_10xv1_2|Sig|0.0841|0.0488|
> ||||pkVI|0.1286|0.1961|
> ||||SAT2|0.3548|0.1565|
> |Bone|batch_27|batch_26|Sig|0.2665|0.2477|
> ||||pkVI|0.2107|0.1939|
> ||||SAT2|0.4897|0.3525|
> |Bone|batch_43|batch_44|Sig|0.1597|0.1764|
> ||||pkVI|0.2118|0.1795|
> ||||SAT2|0.4479|0.4540|
> |Cortex|batch_2|batch_1|Sig|0.0240|0.0216|
> ||||pkVI|0.0973|0.0717|
> ||||SAT2|0.4007|0.2635|
> |Cortex|batch_3|batch_2|Sig|0.4930|0.3629|
> ||||pkVI|0.5145|0.2610|
> ||||SAT2|0.0880|0.0843|
> |Retina|D19D003|D018_13|Sig|0.0353|0.1037|
> ||||pkVI|0.0935|0.0763|
> ||||SAT2|0.0000|0.0000|
> |Retina|D021_13|D19D003|Sig|0.8088|0.5642|
> ||||pkVI|0.0267|0.0393|
> ||||SAT2|0.3032|0.2500|
>
> 5. Clarification on non-pretrained ChromFound:
>
>    In manuscript, we have already included non-pretrained ChromFound as a baseline in Fig. 3 and Table 3. We emphasize that for tasks such as cell clustering and denoising, ChromFound is evaluated in zero-shot settings. Non-pretrained ChromFound loading randomly initialized weights in these scenarios is not a meaningful or fair baseline. We hope the reviewer understands our rationale, and we are happy to further discuss this point if needed.
>
> ## [2]. Other peak calling methods and VAE pretraining
> We sincerely thank the reviewer for insightful suggestions. In response, we perform an experiment to directly compare ChromFound with VAE-based models trained on reference-peak-aligned data. Specifically, we adopt the cPeaks reference set proposed by the recent work "A generic reference defined by consensus peaks for scATAC-seq data analysis", which defines 1,657,194 peaks across the human genome. We use a 20,000-cell PBMC subset (same as in Table 4, Row 6) as the training data. After mapping peaks to the cPeaks reference and filtering out peaks and cells with all-zero values, we obtain a training set of 306,784 peaks and approximately 18,000 cells.
>
> We train four VAE-based baselines on the aligned data and compare their performance on zero-shot cell clustering (Table 1) against ChromFound trained on the same aligned data (Ours). Evaluation metrics and datasets follow Table 1, all aligned to 306,784 peaks. Results are summarized below:
>
> |Dataset|Model|ARI|FMI|NMI|AMI|
> |-|-|-|-|-|-|
> |Cortex(batch 1)|SCALE|0.2932|0.5050|0.4127|0.4081|
> ||SCALEX|0.4230|0.6078|0.5543|0.5513|
> ||pkVI|0.3690|0.5643|0.5023|0.4983|
> ||psVI|0.3364|0.5408|0.4636|0.4594|
> ||Ours|0.5858|0.7244|0.7519|0.7499|
> |Cortex(batch 2)|SCALE|0.2484|0.4052|0.3657|0.3609|
> ||SCALEX|0.4481|0.5679|0.5742|0.5710|
> ||pkVI|0.3610|0.5035|0.4876|0.4835|
> ||psVI|0.2820|0.4374|0.4104|0.4058|
> ||Ours|0.5751|0.6720|0.7145|0.7123|
> |Heart(av 3)|SCALE|0.3543|0.4817|0.5228|0.5195|
> ||SCALEX|0.2795|0.4150|0.4787|0.4752|
> ||pkVI|0.3023|0.4437|0.4816|0.4779|
> ||psVI|0.3778|0.5025|0.5446|0.5414|
> ||Ours|0.5400|0.6347|0.6588|0.6567|
> |Heart(av 10)|SCALE|0.4051|0.6506|0.4461|0.4453|
> ||SCALEX|0.3013|0.5685|0.4199|0.4191|
> ||pkVI|0.3966|0.6080|0.5018|0.5011|
> ||psVI|0.3939|0.5997|0.5010|0.5003|
> ||Ours|0.6150|0.7695|0.6968|0.6963|
> |Retina(D026_13)|SCALE|0.3065|0.5771|0.4447|0.4440|
> ||SCALEX|0.2430|0.5149|0.4127|0.4117|
> ||pkVI|0.2514|0.5260|0.4651|0.4644|
> ||psVI|0.2347|0.5122|0.4063|0.4055|
> ||Ours|0.4888|0.7081|0.7241|0.7238|
> |Retina(D19D008)|SCALE|0.3143|0.5092|0.4220|0.4210|
> ||SCALEX|0.2490|0.4542|0.3920|0.3909|
> ||pkVI|0.2664|0.4693|0.4745|0.4736|
> ||psVI|0.2396|0.4473|0.3937|0.3926|
> ||Ours|0.5368|0.6839|0.7664|0.7660|
> |PBMC(VIB_10xv1_1)|SCALE|0.5769|0.6756|0.7066|0.7057|
> ||SCALEX|0.5718|0.6514|0.7205|0.7196|
> ||pkVI|0.5306|0.6249|0.7057|0.7046|
> ||psVI|0.5593|0.6479|0.6742|0.6732|
> ||Ours|0.5999|0.6827|0.7434|0.7424|
> |PBMC(BIO_ddseq_1)|SCALE|0.4031|0.5797|0.5380|0.5374|
> ||SCALEX|0.3326|0.5430|0.5465|0.5458|
> ||pkVI|0.3830|0.5804|0.5737|0.5731|
> ||psVI|0.3959|0.5917|0.5509|0.5503|
> ||Ours|0.4347|0.6237|0.5867|0.5861|
>
> We highlight two key observations from this experiment:
> 1. Baseline models perform poorly on unseen tissues. We find that only 30% of peaks in retina, 35% in cortex, and 70% in heart overlap with the cPeaks-aligned PBMC training peaks, suggesting that cross-tissue peak heterogeneity is a major factor limiting generalization.
> 2. Using cPeaks reduce ChromFound's performance compared to its native OCRs, suggesting that reference-based harmonization cannot capture novel or shifted OCRs introduced by batch effects or disease-specific variation. In contrast, ChromFound’s dynamic OCR tokenization allows the model to implicitly learn genomic distance and neighborhood structures, enhancing both robustness and extensibility.
>
> ## [3]. Minor questions
> 1. _OCR filtering when cell type labels are unavailable_
>
>    When cell labels are unavailable, we apply TF-IDF + LSI to infer pseudo labels for OCR filtering. We will clarify this in the manuscript.
> 2. _Choice of loss function_
>
>    Due to the high sparsity of scATAC-seq data, we adopt a dual masking strategy (Sec 3.3) during pretraining, masking equal number of both nonzero and zero peaks. Under this setting, Poisson/ZINB losses often cause unstable optimization due to gradients dominated by nonzero entries. Instead, MSE loss on log-transformed, normalized signals ensures stability. This choice is also supported by xTrimoGene discussed in both their manuscript and rebuttal.
> 3. _Ground truth in Fig. 4_
>
>    We would like to clarify that the ground truth labels in Fig. 4 are not ABC predictions, but experimental CRISPRi perturbation data in the Fulco4K dataset, as stated in lines 235–236 of the manuscript. Additionally, the citation of Fulco4K dataset will be corrected as "Activity-by-contact model of enhancer–promoter regulation from thousands of CRISPR perturbations".
> 4. _Ablation study on the Mamba layer_
>
>    Thank you for the comment. To clarify, Question 3 in our ablation study is designed to assess the importance of long input peak sequences, not the effectiveness of the Mamba layers. Many existing scATAC-seq methods apply dimensionality reduction (e.g., highly variable peaks or LSI), but we aim to show that modeling long-range chromatin context leads to better cell representations and cross-omics performance.
>
>    The effectiveness of the Mamba layers is directly evaluated in Question 1. The significant performance drop in this setting highlights the contribution of each component, especially Mamba layer.
>
> ## Conclusion
> We sincerely thank the reviewer for these thoughtful and professional suggestions. We hope our responses have addressed the reviewer’s concerns. Please let us know if further clarification is needed.

---

> > ### Comment · Reviewer_dMbP · 2025-08-07
> > **Response to author rebuttals**
> >
> > Many thanks to the authors for performing these additional experiments in such a short period of time. The results are quite impressive, though at points unintuitive (not necessarily wrong, just surprising given my previous usage of these methods). I have a few additional questions (apologies if they're already included in the appendix or your response and I missed them) that might help me build some intuition for these results:
> >
> > 1. Let's focus on a specific evaluation, Cortex (batch 1) for cell clustering in experiment [1] of your response. My understanding is the same peaks are passed to all methods. How many peaks are there? How many cells? I find it surprising that VAEs do so poorly on a simple cell clustering task unless there are very few cells (so pre-training becomes essential) or too many peaks (such that they can't handle the high-dimensionality well).
> >
> > 2. It's still unclear to me if the baselines are run with reasonable hyperparameters. I am most familiar with peakVI/poissonVI, which is why I will focus on those methods. How many training epochs were they run for/what learning rate/is early stopping used?
> >
> > 3. I am not sure why it is unfair for non-pretrained ChromFound to be included in Table 1. I think there should be three versions of ChromFound evaluated: (1) pre-trained and fine-tuned on the dataset of interest, (2) only pre-trained, and (3) only trained on on the dataset of interest.
> >
> > 4. Thank you for running experiment [2]. These results seems to indicate that VAE models pre-trained on a small amount of PBMC data and applied zero-shot to tissues as different as retina and the cortex do poorly. This makes perfect sense. Somehow ChromFound does a lot better. So well in fact that VAEs trained specifically on the dataset of interest (taking numbers from experiment [1]) do worse than ChromFound pre-trained on PBMCs and applied zero shot to this very different tissue. Is that a correction interpretation of the results? If so, can you provide some intuition as to how ChromFound has such impressive generalization capabilities? I understand that OCR tokenization helps and can allow the model to treat peaks in the test set similar to close by peaks in the pre-training set, but I am shocked it helps that much.
> >
> > 5. The cell type annotation results of baselines are shockingly poor. But I think this is because of batch effects between the training and test sets. Do you use something like scArches do integrate the test (query) dataset into the training (reference) dataset?
> >
> > 6. I'm still quite confused how exactly the ablations for Question 3 are run. How are the number of OCRs per cell reduced?

---

> ### Author Response · Authors · 2025-08-04
> **We Appreciate the Feedback and Welcome Further Discussion**
>
> Dear reviewer:
>
> We sincerely thank you for the time, expertise, and thoughtful suggestions provided throughout the review process. In this rebuttal, we have done our best to carefully address every concern raised, including expanded benchmarking, detailed comparisons with alternative peak harmonization strategies, and clarification of modeling and implementation details. These revisions were made with the goal of strengthening the work and making our contributions clearer.
>
> We greatly value your feedback and would be very glad to further discuss any remaining questions or concerns. Please do not hesitate to reach out. We truly welcome continued dialogue and are committed to engaging with all comments in depth. Thank you again for your time, insight, and consideration.

---

> ### Author Response · Authors · 2025-08-08
> **Reply to Official Comment by Reviewer dMbP (1/2)**
>
> We sincerely thank the reviewer for the follow-up questions and for acknowledging the additional experiments. We address each point below.
>
> ## [Q1] Details of evaluation datasets
> We appreciate your insightful questions. Below we provide detailed information about the evaluation datasets in experiment [1]:
>
> |Dataset|#Cell|#Peak|
> |-|-|-|
> |Cortex(batch 1)|1,119|141,389|
> |Cortex(batch 2)|1,416|212,793|
> |Heart(av 3)|3,733|333,393|
> |Heart(av 10)|2,785|343,027|
> |Retina(D026_13)|7,529|235,366|
> |Retina(D19D008)|4,575|232,354|
> |PBMC(BIO_ddseq_1)|5,592|192,229|
> |PBMC(VIB_10xv1_1)|2,707|363,066|
>
> ## [Q2] Hyperparameters of peakVI/poissonVI
> We thank the reviewer for pointing out the hyperparameter settings of peakVI and poissonVI. We follow the default setting in the `scvi-tools` library. Specifically we detail the hyperparameters you mentioned below:
>
> |Model|training epochs|learning rate|early stopping patience|
> |-|-|-|-|
> |peakVI|500|0.0001|50|
> |poissonVI|500|0.0001|50|
>
> We observe that peakVI typically runs through 500 epochs, whereas poissonVI usually stops after fewer than 100 epochs due to the early stopping strategy. Thank your again for your professional suggestion. We hope this clarification of the hyperparameter settings will address your question.
>
> ## [Q3] Non-pretrained ChromFound
> Thanks for your suggestions and clarification. We include additional results as below, including pre-trained and continuous training on the dataset of interest (ChromFound with continuous training on evaluation datasets) and training from scratch on the dataset of interest (ChromFound trained from scratch on evaluation datasets).
>
> |Dataset|Model|ARI|FMI|NMI|AMI|
> |-|-|-|-|-|-|
> |Cortex(batch 2)|ChromFound with continuous training on evaluation datasets|0.6306|0.7176|0.7306|0.7285|
> ||ChromFound trained from scratch on evaluation datasets|0.6100|0.6997|0.7292|0.7271|
> ||ChromFound|0.6278|0.7156|0.7321|0.7299|
> |Retina(D19D008)|ChromFound with continuous training on evaluation datasets|0.6710|0.7783|0.8189|0.8186|
> ||ChromFound trained from scratch on evaluation datasets|0.6362|0.7534|0.8038|0.8035|
> ||ChromFound|0.6688|0.7767|0.8183|0.8179|
>
> As shown in the table above, ChromFound with continuous training on evaluation datasets achieves better performance than ChromFound and ChromFound trained from scratch on evaluation datasets. Nevertheless, training on the evaluation datasets incurs significant additional computational cost. We report the inference and training speed of one A100 80G GPU, along with GPU memory usage.
>
> |Dataset|#Cell|#Peak|Setting|Batch Size|Total time(s)|GPU memory (GB)|
> |-|-|-|-|-|-|-|
> |Cortex(batch 2)|1416|212,793|Training|8|2292|63.53|
> ||||Inference|8|336|13.9|
> |Retina(D19D008)|4575|232,354|Training|8|8270|68.31|
> ||||Inference|8|1090|15.12|
>
> Given the additional computational cost and the limited performance gain, we recommend generating cell representations in a zero-shot setting without further training.
> Due to time constraints, we will include results for all datasets in the revised manuscript, along with corresponding computational resource requirements, as part of the usage instructions for ChromFound.
> We once again thank the reviewer for the valuable suggestion.

---

> ### Author Response · Authors · 2025-08-08
> **Reply to Official Comment by Reviewer dMbP (2/2)**
>
> ## [Q4] Correction interpretation of the results
> We sincerely thank the reviewer for pointing out the correct interpretation of the results. As you mentioned that "VAEs may do poorly on very few cells or too many peaks", we also notice that the used peak number of datasets in the tutorial of peakVI implemented in scvi-tools is 33142, which is much smaller than the number of peaks in our datasets. Therefore, we believe the poor performance of peakVI/poissonVI in our experiments is likely due to the large number of peaks in our datasets.
> We include the additional experiments to filter peaks with highly variable selection and retrain peakVI/poissonVI on the filtered datasets. The results are summarized below:
>
> |Dataset|Model|#Peak|ARI|FMI|NMI|AMI|
> |-|-|-|--|-|-|-|
> |Cortex(batch 2)|peakVI|212,793|0.5512|0.6530|0.7050|0.7028|
> ||peakVI(hv)|30,000|0.5867|0.6814|0.7221|0.7200|
> ||poissonVI|212,793|0.5334|0.6388|0.6004|0.5983|
> ||poissonVI(hv)|30,000|0.5723|0.6703|0.7203|0.7181|
> ||ChromFound trained on PBMC cPeaks|212,793|0.5751|0.6720|0.7145|0.7123|
> ||ChromFound|212,793|0.6278|0.7156|0.7321|0.7299|
> |Retina(D19D008)|peakVI|232,354|0.4702|0.6318	|0.7229|0.7224|
> ||peak(hv)|30,000|0.5907|0.7395|0.7841|0.7836|
> ||poissonVI|232,354|0.4978|0.6527|0.7400|0.7395|
> ||poissonVI(hv)|30,000|0.6037|0.7443|0.7966|0.7961|
> ||ChromFound trained on PBMC cPeaks|232,354|0.5368|0.6839|0.7664|0.7660|
> ||ChromFound|232,354|0.6688|0.7767|0.8183|0.8179|
>
> As shown in the table above, peakVI and poissonVI perform much better after highly variable peak selection, confirming their strength as baselines under reduced dimensionality.
> However, as the input dimensionality increases, their performance exhibits a marked decline, suggesting they may struggle to model long-range chromatin dependencies.
> ChromFound’s hybrid WPSA-Mamba architecture addresses this limitation by integrating local _cis_-regulatory context with efficient long-range modeling, enabling broader genome-wide applications such as enhancer–gene link inference and perturbation effect prediction (Section 4.3).
>
> Due to time constraints, we are unable to extend this integration experiment to all datasets. We will be glad to include results for all datasets and additional baselines with highly variable peak selection in the revised manuscript. Thank you again for your professional suggestion.
>
> ## [Q5] Batch effect in the cell type annotation
> We sincerely thank the reviewer for this professional and insightful suggestion. We did not integrate the test (query) dataset into the training (reference) dataset in the response before. Following your professional suggestion, we implement scArches for scPoli to compare cell type annotation performance with and without applying scArches-based integration between the training (reference) and test (query) datasets. The integrated setting yields a substantial performance improvement, fully consistent with your observation that batch effects largely account for the poor baseline results.
>
> |Dataset|Train|Test|Method|Accuracy|F1|
> |-|-|-|-|-|-|
> |Bone|batch_43|batch_44|scPoli|0.2500|0.1822|
> ||||**scArches**-integrated scPoli|0.8137|0.8005|
> ||||ChromFound|0.8368|0.8335|
> |Cortex|batch_2|batch_1|scPoli|0.4120|0.2286|
> ||||**scArches**-integrated scPoli|0.8640|0.6973|
> ||||ChromFound|0.9366|0.7715|
>
> Due to time constraints, we are unable to extend this integration experiment to all datasets. We will be glad to include results for all datasets and additional baselines with train/test integration by scArches in the revised manuscript. We greatly appreciate your suggestion, which has helped us optimize and strengthen our work.
>
> ## [Q6] The number of OCRs per cell reduced in ablation study Question 3
> We sincerely thank the reviewer for the follow-up questions. For pretraining, we set the maximum OCR sequence length to 440,000. If a dataset contains more than 440,000 OCRs, we retain the top 440,000 most variable OCRs; if it contains fewer, we apply zero padding to match the maximum length. In the ablation for Question 3, we compare maximum lengths of 220,000 and 110,000 OCRs. The reduction in OCRs for these settings is performed via dataset-specific highly variable OCR selection, ensuring that the retained peaks capture the most informative variability within each dataset.
>
> ## Summary
> We are deeply grateful to the reviewer for their thoughtful and professional feedback. Your comments not only demonstrate a thorough understanding of our work, but also reflect a high level of expertise and academic rigor.
> Your insightful suggestions significantly improve the clarity and depth of our manuscript. We feel privileged to have our work evaluated by someone with such discernment and scholarly acumen.
> We hope that our response has addressed all your concerns. Thank you again for your invaluable contribution to the refinement of our research.

---

### Author Response · Authors · 2025-08-09
**Rebuttal Summary for ChromFound**

In this paper, we present ChromFound, a foundation model for scATAC-seq data that integrates genome-aware tokenization, specialized long-range modeling, and robust cross-tissue generalization.
We sincerely thank the AC and all reviewers for your constructive engagement, and are encouraged by the strong recognition of our work, including one Strong Accept and another score raised to Accept. In response, we add extensive experiments, expanded benchmarking, hyperparameter sensitivity analysis, and methodological clarifications, further strengthening ChromFound's technical soundness, clarity, and impact.

We summarize the key points of our responses during rebuttal and discussion:
1. Comprehensive baseline expansion
   - Signac, SnapATAC2, peakVI, and poissonVI are included across clustering, batch-effect correction, and annotation tasks for fair and widely recognized comparisons. (requested by reviewer dMbp)
   - Shallow baselines, including logistic regression and a two-layer MLP, confirms ChromFound’s gains even over strong alternatives. (requested by reviewer dMbp and VAFg)
   - Evaluations on continuous training and training-from-scratch variants of ChromFound confirm its consistent superiority. (requested by reviewer dMbp)
   - We optimize peakVI/poissonVI by highly variable peak selection, beneficial in low-dimensional settings but degraded with full OCR lengths, highlighting ChromFound’s ability to model long-range chromatin dependencies without accuracy loss. (requested by reviewer dMbp)
2. Harmonization and peak-calling methods comparison:
   - Compared with a pre-defined OCR reference (cPeaks), dynamic OCR tokenization outperforms on unseen tissues, showing robustness of genome-aware OCR representation. (in response to reviewer dMbp and VAFg)
   - Aligned-data experiments with VAE-based models confirm generalization mainly stems from unified OCR tokenization. (in response to reviewer dMbp)
   - Downsampling experiments in Fig. 2, along with strategies such as log-transformation, dataset-specific normalization, genome-aware reconstruction, and large heterogeneous pretraining, jointly support robust cross-tissue modeling and broader applicability over pre-defined OCR references. (in response to dMbp, ybr7 and VAFg)
3. Architectural validation and hyperparameter sensitivity analysis:
   - Each component of ChromFound is tailored to scATAC-seq for biologically meaningful representation and practical applications. (in response to reviewer VAFg)
   - Transformer-based proxies (Geneformer, scGPT) saturate well below ChromFound’s accuracy on full-length OCR inputs, indicating that its architecture, rather than data scale alone, enables effective modeling of ultra-long genomic sequences. (in response to reviewer ybr7)
   - WPSA outperforms linear attention variants (Performer, Linformer) in balancing scalability and representation. (in response to reviewer VAFg)
   - Hyperparameter sensitivity experiments on positional embedding temperature $temp$ and Mamba projection dimension $D_{low}$ identifies optimal settings that balance accuracy, memory efficiency, and computational cost. (in response to reviewer ybr7)
4. Loss function design and sparsity handling
   - We emphasize that the balanced masking strategy prevents trivial zero-prediction bias, and that MSE loss on log-normalized accessibility values avoids the instability of Poisson/ZINB loss across diversified datasets, both ensuring robustness to zero and non-zero OCRs imbalance. (in response to reviewer ybr7, VAFg and dMbp)
   - We highlight that ChromFound captures population-level statistical patterns from diverse datasets, enabling it to implicitly distinguish consistent regulatory signals from dataset-specific sparsity or noise. (in response to reviewer VAFg)
5. Robustness to biological diversity
   - Strong performance on Alzheimer’s and myocardial infarction datasets validates ChromFound’s applicability to diverse disease contexts beyond healthy tissues. (in response to reviewer ybr7)
   - We explain the retina-specific performance improvement under more aggressive downsampling in Figure 2 by showing that retina datasets contain a higher proportion of high-frequency OCRs, where downsampling reduces redundancy and improves signal-to-noise ratio. (in response to reviewer VAFg)
6. Practical application
   - Training and inference runtime as well as memory usage on A100 GPUs provides transparent resource requirements for reproducibility and deployment planning. (in response to dMbp and ybr7)

We sincerely thank the AC for ensuring a fair and constructive process, and all reviewers for their insightful feedback. Your comments have directly strengthened our experiments, refined our methodology, and more clearly demonstrated ChromFound’s robustness, scalability, and broad applicability. We also deeply appreciate your dedication and contributions to advancing the AI for Science community, which have greatly enriched and elevated this review process.

---

### Note · Authors · 2025-08-13

We sincerely thank the AC for overseeing a thorough and constructive process, and all reviewers for your insightful and rigorous feedback.

ChromFound is one of the first foundation model for scATAC-seq, integrating genome-aware tokenization, specialized long-range modeling, and robust cross-tissue generalization. By capturing dynamic regulatory patterns across the genome, it enables biologically meaningful representation and broad multi-omics applications. During the rebuttal, we systematically examined every reviewer concern and provided targeted, evidence-based resolutions:
- Baselines: Expanded coverage with Signac, SnapATAC2, peakVI, poissonVI, and shallow baselines; tested continuous-training and from-scratch variants, consistently confirming superiority.
- Harmonization & peak-calling: Demonstrated that dynamic genome-aware OCR tokenization outperforms large pre-defined references and VAE-based pretrained methods on unseen tissues, showing strong generalization and robustness.
- Architecture: Validated novelty through targeted ablations; outperformed Transformer proxies (Geneformer, scGPT); WPSA achieved a better accuracy–efficiency balance than linear attention variants.
- Loss & sparsity: Balanced masking and MSE loss on log-normalized values avoided instability and bias, while capturing population-level patterns beyond dataset-specific noise.
- Biological diversity: Achieved strong results on Alzheimer’s and myocardial infarction datasets; provided mechanistic insight into retina-specific downsampling gains.
- Practical readiness: Profiled runtime and memory usage for deployment transparency.

We have made every effort to address all concerns raised by incorporating targeted experiments and clarifications. To the best of our knowledge, ChromFound is the first model to achieve zero-shot cross-tissue generalization and introduce a genome-wide framework for gene-enhancer link and perturbation response prediction in real-world scATAC-seq applications, building upon a foundation designed for practical impact. We believe it holds broad interest for the community and merits widespread dissemination. We sincerely hope these updates will be considered in the final decision. Thank you again for your time, expertise, and dedication to advancing rigorous, impactful research in AI for Science.

---

### Decision · Program_Chairs · 2025-09-17

**Decision:**

Accept (poster)

**Comment:**

The submission describes a foundation model for scATAC-seq data which can be pretrained on data from diverse cell types and tissues and used in a zero-shot or fine-tuned fashion on downstream tasks. Reviewers had high appraisal for the novelty and significance for this application domain. Questions about the technical details of the experiments led to a lively discussion, in which authors provided further experimental comparisons. 3/4 reviewers indicated positive appraisal of these new results presented in the rebuttal and the resolution of most concerns.